

# Dynamic Forcing Behind Hurricane Lidia's Rapid Intensification

**Mauricio López-Reyes[1,2,3], María Luisa Martín-Pérez[4,5], Carlos Calvo-Sancho[4,6], Juan Jesús González-Alemán[7]**

[1]Instituto de Astronomía y Meteorología (IAM), Centro Universitario de Ciencias Exactas e Ingenierías (CUCEI), Departamento de Física, Universidad de Guadalajara, Guadalajara, Mexico.

[2]Department of Earth Physics and Astrophysics, Faculty of Physics, Complutense University of Madrid, Madrid, Spain.

[3]Instituto Frontera A.C., Departamento de Investigación, Tijuana, México.

[4]Department of Applied Mathematics, Faculty of Computer Engineering, University of Valladolid, Spain.

[5]Institute of Interdisciplinary Mathematics (IMI), Complutense University of Madrid, Madrid, Spain.

[6]Centro de Investigaciones sobre Desertificación, Consejo Superior de Investigaciones Científicas (CIDE, CSIC-UV-Generalitat Valenciana), Climate, Atmosphere and Ocean Laboratory (Climatoc-Lab), Moncada, Valencia, Spain

[7]Department of Development and Applications, Agencia Estatal de Meteorología (AEMET), Madrid, Spain.

Corresponding author: C. Calvo-Sancho (carlos.calvo.sancho@uva.es)

**Key words**

Rapid intensification, ensemble prediction, tropical cyclone, extratropical interactions

**Key points**

- A mid-to-upper level trough enhanced vertical motion and divergence over Hurricane Lidia, triggering rapid intensification.
- Stronger Trenberth forcing, eddy flux convergence, and vorticity advection were observed in ensemble members that captured RI.
- Ensemble diagnostics revealed that dynamic forcing preceded RI onset, suggesting a causal role beyond thermodynamic conditions.

**Abstract**

This study examines Hurricane Lidia's rapid intensification (RI) in the understudied northeastern Pacific, focusing on its interaction with an upper-level trough. Using IFS-ECMWF ensemble forecasts and ERA5 reanalysis, we analyze the large-scale dynamical mechanisms driving Lidia's intensification. Results show that the trough played a crucial role in promoting RI by enhancing synoptic-scale ascent, upper-level divergence, and eddy flux



convergence. In the higher-intensification ensemble group, stronger Trenberth forcing
emerged prior to RI onset, suggesting a causative role in preconditioning the storm
environment. This dynamical forcing likely triggered latent heat release, which in turn
modified the upper-level potential vorticity structure and contributed to a subsequent
reduction in vertical wind shear. In contrast, the lower-intensification group exhibited
weaker forcing, higher shear, and a lack of sustained ventilation. These findings highlight the
importance of diagnosing early dynamical triggers for RI, particularly in regions where
operational access to high-resolution models is limited. This approach provides a cost-
effective framework for anticipating RI using ensemble-based diagnostics and could serve
as a valuable forecasting tool in data-sparse areas such as the Pacific coast of Mexico. Future
studies should combine this large-scale methodology with high-resolution simulations to
better capture storm-scale processes and validate multi-scale interactions in RI events.



## 1   Introduction

México is among the countries most affected by tropical cyclones (TCs) from both the Atlantic and Pacific Oceans (Larson et al., 2005; López-Reyes et al., 2024). While the Atlantic basin has traditionally garnered more research attention, largely due to the severe economic and social impacts of TCs in the United States, there is a pressing need to expand research efforts in the northeastern Pacific basin, where fewer studies have been conducted (García Franco et al., 2024). In recent years, various major hurricanes, such as Patricia, Lidia, and Otis caused large economic losses and scores of deaths in México (Pasch, 2024; García Franco et al., 2024). These events also posed challenges for numerical weather prediction models, particularly in forecasting their tracks and intensification processes. As highlighted by Shi and Chen (2021), one of the key obstacles is improving the prediction of TCs that undergo rapid intensification (RI), defined as an increase of at least 30 kt ($\approx$ 54 km/h) in maximum sustained wind speed within a 24-h period (Kaplan and DeMaria, 2003). Recent studies have shown a rise in the frequency of RI events in the Atlantic basin, driven primarily by ocean warming (Majumdar et al., 2023; Li et al., 2023). The northeastern Pacific, however, has also experienced extreme intensification rates, with Patricia (2015), Willa (2018), and Otis (2023) ranking among the most rapidly intensifying storms on record. Similarly, during the 2024 hurricane season, Hurricane Milton underwent explosive intensification, posing a significant challenge for intensity forecasting (Pasch, 2024). Enhancing our ability to forecast RI is essential for reducing the risk these powerful storms pose to vulnerable communities and critical infrastructure.

Although RI is strongly influenced by thermodynamic factors, such as high sea surface temperatures (SSTs) and ocean heat content, dynamic factors also play a pivotal role. Interactions between TCs and upper-level troughs have been shown to significantly affect storm intensity (Fischer et al., 2019). According to Avila (1998), Hanley et al. (2001), López-Reyes et al. (2021) and DeMaria et al. (2021), forecasting intensity changes in TCs remains one of the biggest challenges, particularly during RI. The difficulties in forecasting RI stem from the complex factors involved in the occurrence of RI, such as the large-scale environment, internal dynamics and multiscale interactions (Kaplan et al., 2010; Zhang and Chen 2012; Bhalachandran at al., 2020; Wang et al., 2021; Shi and Chen 2021). Over the past few years, there has been notable progress in understanding the internal dynamics that govern RI. As found in Chen et al. (2019) and Shi and Chen (2021), the main dynamical and thermodynamic drivers of TC RI include strong upper-level divergence and strong boundary-layer convergence as well as a weak deep-layer vertical wind shear (VWS), higher relative humidity throughout the vertical column, and high intensification potential (details in Emanuel, 1988) associated with SST.  Other studies have highlighted the significance of the deep convective region surrounding the eyewall and the large convergence of angular momentum into TC (Stevenson et al., 2014; Komaromi and Doyle 2018; Ryglicki et al., 2021). Furthermore, research studies have identified a relevant relationship between the structure and size of TCs, their environmental conditions, and the RI rate (Carrasco et al., 2014; Shi and Chen 2019; Tao et al., 2022; Ston et al., 2023; Nayaranan et al., 2024).





Since the general conditions favoring TC RI are well-known, other factors may influence
the overall intensification procesess. In a follow-up theoretical study, Lenoux et al. (2016)
identified an optimal TC-trough alignment that promotes interaction (Komaromi and Doyle
2018; Shato et al., 2020). Similarly, studies by  DeMaria et al. (1993), Hanley et al. (2001),
and Peirano et al. (2016) suggest that an approaching upper-tropospheric trough can play a
critical role in  hurricane intensification. However, trough interactions can also limit TC
intensification, depending on the configuration of the trough and its associated jet stream.
For example, increased dry air entrainment or increased VWS can inhibit TC development
(Peirano et al., 2016). Recent research has identified specific synoptic configurations that
favor RI, including short zonal wavelengths and favorable upstream displacements between
the TC and the trough (Fischer et al., 2019). Qiu et al. (2020) also showed how important
eddy flux convergence (EFC) is for making TC-trough interactions, particularly when large-
scale circulation patterns favor stronger upper-level divergence. In the same way, Yan et al.
(2021) found that upper-tropospheric cold lows could enhance EFC, reduce inertial stability,
and strengthen upper-level divergence, leading to RI. These studies show that TC-trough
interactions can have two effects and stress how important is to figure out what
environmental conditions make TC stronger or weaker.
Although TC–trough interactions have been extensively studied in the Atlantic basin, the
northeastern Pacific remains understudied. During El Niño events, the subtropical Pacific
warms considerably, increasing the likelihood of interactions between TCs and the jet
stream, increasing the likelihood of dynamic interactions (Luna-Niño et al., 2021; Ling et al.,
2024). In the Atlantic basin, on the other hand, these interactions often occur at higher
latitudes over less populated areas in the Atlantic basin. In contrast, TCs in the northeastern
Pacific tend to curve toward land, which puts densely populated areas in Mexico at risk. The
fact that warm SSTs and the jet stream interact during El Niño events shows how important
it is to do focused research in this area.
Additionally, the proximity of northeastern Pacific TCs to mountainous terrain introduces
further challenges for forecasting (DiMego et al., 1976). The most intense hurricanes that
have affected Mexico typically occur during late summer and early autumn, as was the case
with Hurricane Lidia in mid-October 2023. The devastating case of Hurricane Otis in October
2023, in which all global models failed to capture its RI, underscored the urgent need for
improved understanding of RI processes in this region. This event caused dozens of fatalities
and severe and widespread damages in Acapulco, highlighting Mexico's vulnerability to such
phenomena and the critical need for better forecast capabilities (Emanuel, 2024; Servicio
Meteorológico Nacional, 2023). Given the high SSTs during this season, trough-TC
interaction is particularly relevant during October and November, as many TCs turn
eastward during this period. This turning is  influenced by the subtropical jet stream,
typically positioned between 25°N and 35°N, especially during seasons when El Niño events
are present (Luna-Niño et al., 2021; Tong et al., 2023). The jet stream may enhance
interactions between midlatitude troughs TCs; however, to the authors' knowledge, no
previous studies have specifically investigated the role of the jet stream in TC RI in the
northeastern Pacific. In contrast to most prior studies, which primarily focus on



thermodynamic drivers in the North Atlantic basin, our research emphasizes the dynamic forcing mechanisms associated with trough-TC interactions, such as quasi-geostrophic ascent, EFC, and enhanced upper-level divergence in the intensification of Hurricane Lidia. By analyzing ensemble prediction system (EPS) outputs and ERA5 reanalysis data, we provide a comprehensive assessment of the conditions that favored Lidia's RI, offering novel insights into the dynamics of TC intensification in the northeastern Pacific.

Moreover, the northeastern Pacific lacks operational mesoscale models, making reliable predictions of RI particularly challenging. In this context, EPSs have emerged as a valuable tool for operational forecasting, providing insights into uncertainty and enabling the evaluation of potential risk scenarios several days in advance. Using EPS outputs, this study seeks to address these challenges and improve our understanding of the conditions that favor RI in the northeastern Pacific.

This paper is organized as follows: Section 2 presents the data models and methods, including the ensemble configurations and diagnostics tools. Section 3 describes the synoptic conditions that influenced Hurricane Lidia's RI and the main dynamical processes involved. Finally, Section 4 provides a summary of findings and concluding remarks.

## 2   Data Models and Methods

The data sets are based on forecasts from the Integrated Forecasting System (IFS) of the European Centre for Medium-Range Weather Forecasts (ECMWF). This study uses the operational perturbed forecast ensemble  generated by the EPS (Cycle 48r1: ECMWF, 2023) with 50 perturbed members is used. Each perturbed member has a horizontal resolution of 0.1° and 137 vertical levels. The last initialization is selected since it features a large spread of Hurricane Lidia trajectories, corresponding to the October 8th, 00:00 UTC initialization, with 1-hour time steps during 96 h. Additionally, to assess the performance of each composite group, key atmospheric fields are computed using data from the ERA5 climate reanalysis (Hersbach et al., 2020) with 0.25° horizontal resolution and 37 vertical levels, during the period with the highest intensification rate.

Several dynamic and thermodynamic variables were utilized in this study, such as mean sea level pressure ($MSLP$), temperature ($T$), geopotential height ($Z$), zonal and meridional wind components ($u, v$), potential temperature ($\theta$), SST and relative humidity ($RH$). See Table 1 for additional details. To evaluate the role of the trough in the trajectory and intensity change of Hurricane Lidia. Trajectories and intensification rate for all members are determined using MSLP and divided them into two intensification rate groups (IRG) based on the $P_{20}$ (lower intensification rate) and $P_{80}$ (highest intensification rate) percentiles of MSLP. In addition, members who meet the RI definition are identified using wind threshold criteria (greater than 54 km/h in 24-h). The NHC best track and official intensity data were used for comparison with both groups. As is common in studies of this nature (Chen et al., 2019; Chen et al., 2021; Hu and Zou, 2021; Collins et al., 2022), synoptic and storm-centered composites (SCC) are derived for the specified fields within a circular area with an 8° radius.



**Table 1.** Details on the atmospheric variables used.

| Variable | Symbol | Pressure levels (PVU) | Units |
|---|---|---|---|
| Mean sea level pressure | $MSLP$ | Surface | $hPa$ |
| Temperature | $T$ | Surface, 1000, 925, 850, 700, 500, 400, 300, 250, 200 | $K$ |
| Geopotential height | $Z$ | 1000, 925, 850, 700, 500, 400, 300, 250, 200 | $m$ |
| Zonal wind component | $u$ | 850, 300, 200 | $ms^{-1}$ |
| Meridional wind component | $v$ | 850, 300, 200 | $ms^{-1}$ |
| Potential temperature | $\theta$ | (1.5-PVU) | $K$ |
| Sea surface temperature | $SST$ | Surface | $K$ |
| Relative humidity | $RH$ | 500 hPa | % |

Based on the previous variables, some derived fields related with TC intensity change (Chen et al., 2021; Mei and Yu 2016) are also computed: VWS calculated between 850 and 200 hPa, irrotational wind ($\vec{V}_{irr}$) at 200 hPa, based on Helmholtz decomposition (details in Chorin et al., 1990 and Cao et al., 2014) and vorticity advection, $\vec{V} \cdot \nabla(\vec{\xi} + f)$ at 500 hPa, were $\vec{\xi} = \frac{\partial v}{\partial x} - \frac{\partial u}{\partial y}$ and $f$ is the planetary vorticity.

Following Bister and Emanuel (1998) and Gilford (2021), potential intensity ($PI$) is also calculated as

$$PI = V_{max} = \left[ \frac{C_k}{C_D} \frac{(T_s - T_0)}{T_0} (h_s^* - h^*) \right]^{1/2},$$

where $C_k$ is the enthalpy surface exchange coefficient, $C_D$ is the momentum surface exchange coefficient, $h_s^*$ is the saturation moist static energy at the sea surface, $h^*$ is the saturation moist static energy of the air above the boundary layer, following to Wing et al. (2015), evaluated at 500-600 hPa. As mentioned in Gilford (2021), tropical cyclone thermodynamic disequilibrium and efficiency were represented by terms $(h_0^* - h^*)$ and $\frac{T_s - T_0}{T_0}$, where $T_s$ is the sea surface temperature and $T_0$ is the outflow temperature level.

To identify regions that favor ascending air movements driven by synoptic-scale dynamical forcing associated with extratropical systems (Loughe et al., 1995; Hanley et al., 2001), ageostrophic wind ($\vec{V}_{ag}$) and its divergence ($\nabla \cdot \vec{V}_{ag}$) are additionally computed. The quasi-geostrophic (QG) omega equation is also used to identify the synoptic ascent flow via Trenberth form (Billingsley, 1998; Bracken and Bosart, 2000). Trenberth QG forcing ($Q$) is calculated using the following expression

$$Q = \left( \sigma \nabla_p^2 + f_0^2 \frac{\partial^2}{\partial p^2} \right) \omega \approx 2 \left[ f_0 \frac{\partial \vec{V}_g}{\partial p} \cdot \nabla \left( \frac{\partial v}{\partial x} - \frac{\partial u}{\partial y} + f \right) \right], \qquad (1)$$





i.e., vertical air movements are proportional to advection by vorticity by thermal wind.
Herein, $\sigma$ is the stability parameter, $f_0$ the Coriolis parameter, $\omega$ the vertical component of
wind ($Pa \cdot s^{-1}$), $\vec{V}_g$ the geostrophic wind vector ($m \, s^{-1}$), $p$ the pressure ($Pa$). Finally, with
the aim of measuring the degree of interaction between the trough and the TC, and
following previous studies in the Atlantic Ocean (Molinari and Vollaro, 1990; Hanley et al.,
2000; Komaromi and Doyle, 2018), the eddy flux convergence (EFC) is defined as
$$EFC = -\frac{1}{r^2}\frac{\partial}{\partial r}\left(r^2 \overline{v_r' v_t'}\right), \tag{2}$$
where $v_r'$ is the perturbation radial wind, $v_t'$ the perturbation tangential wind, and the
overbar denotes the azimuthal mean, computed in storm-relative coordinates. Based on the
methodology of DeMaria et al. (1993) and Hanley et al. (2001), the EFC is computed over a
radial range of 300 to 600 km for each time step during RI period.
To compare atmospheric fields between the most and least intensifying groups,
averages and standard deviations (STD) are calculated, and ensemble difference spatial
distributions ($P_{80} - P_{20}$) are generated to visualize the contrasts between the two groups.
In addition, time series of means and STDs of the thermodynamic and dynamic variables
analyzed between the groups during the simulation period are performed. Finally, a Mann-
Whitney U test is performed to identify regions with statistically significant differences at
the 95% confidence level (Mann and Whitney, 1947).
**3 Results**
3.1 Trajectory and intensity forecast analysis.
Hurricane Lidia originated from a tropical wave on 3 October 2023 (Pasch, 2024).
Between 3 and 5 October, it remained a disorganized system, marked by significant
uncertainty in both track and intensity forecasts (Figures S1). From 5 to 7 October, Lidia
generally tracked westward under the influence of a mid-level ridge but remained poorly
organized. By 8 October, the subtropical jet stream was positioned between 20° and 30°N,
aligned with Lidia's latitude. At this stage, a mid-to-upper-level trough approaching the Baja
California Peninsula began to influence Lidia's motion, steering the system northward and
subsequently eastward.
At approximately 18:00 UTC on 9 October, Lidia entered a phase of intensification
(Pasch, 2024). This intensification was accompanied by a northeastward turn induced by an
approaching trough from the northwest, although considerable spread in forecast
trajectories persisted at this time (Fig. 1a). On 10 October, Lidia underwent RI, with
maximum sustained winds increasing by 82 $km \, h^{-1}$ over an 18-hour period, ultimately
reaching a peak intensity of nearly 220 $km \, h^{-1}$. This placed Lidia at Category 4 on the Saffir–
Simpson Hurricane Wind Scale.



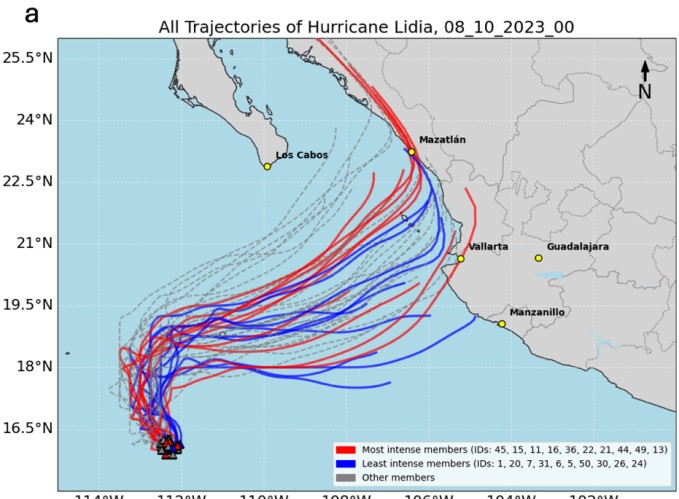

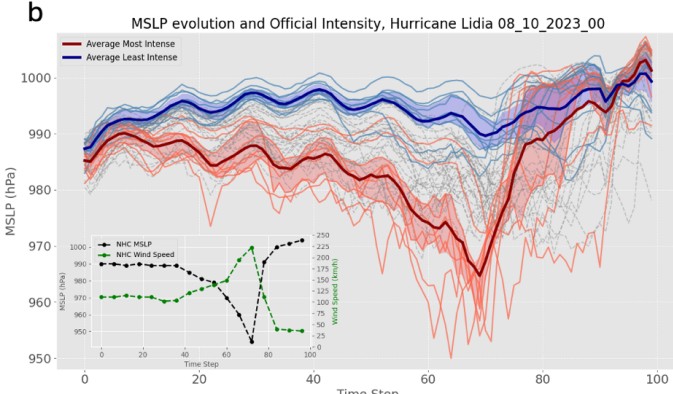

Figure 1. (a) Lidia's Trajectories for all members, highest (lower) IRG in red (blue) line and best track of NHC (black line). (b) Intensity temporal evolution for all members (MSLP), highest (lower) IRG in red (blue) line, shaded areas correspond with interquartile range; real wind speed in green dotted line and MSLP in black dotted line.

Figure 1a shows the trajectories of Hurricane Lidia's ensemble members from the ECMWF, initialized at 00:00 UTC on 8 October. The trajectories of the most intense members are positioned further north relative to those of the lower-intensity members, relative to the NHC best track. This demonstrates that the trough's proximity influenced event predictability, increasing uncertainty in both track and intensity forecasts. This pattern aligns with findings from Ito and Wu (2013), Callaghan (2020), and Sato et al. (2020) in the Atlantic basin, indicating the contribution of synoptic environment to the low predictability of both trajectory and intensity of the cyclone, as here evidenced by the large spread in Figures 1a, b. Furthermore, based on Lidia's MSLP and wind speed temporal evolution (Fig. 1b), we observe that seven members corresponding to the $P_{80}$-ensemble, along with its





mean, successfully simulates Lidia's RI (Figure 1b). Although the simulated intensification is
slightly weaker, the timing is consistent with the actual RI period.

As is well known and formulated in the Emanuel model (Emanuel, 2002) and used in
Chen et al. (2021), the oceanic and atmospheric variables such as $T_s$, $T_0$ and saturation
parameters (eq. 1), determine the PI that the TC could acquire. Both ensembles display
similar PI distributions around Hurricane Lidia (Figs. 2a, b). However, somewhat
unexpectedly the $P_{20}$-ensemble shows a higher PI value ($\approx 250\ km\ h^{-1}$) compared to the
$P_{80}$-ensemble ($\approx 240\ km\ h^{-1}$), although the differences are not statistically significant (not
shown).

Based on the PI time series (Fig. 2c), this diagnostic variable alone does not appear to
support Lidia's RI. Therefore, this suggests that thermodynamic factors are necessary but
not sufficient to trigger RI. This finding is consistent with recent studies (e.g., Gilford, 2021;
Shi and Chen, 2021) which suggest that while PI provides an upper bound, the actual
intensification process is modulated by environmental dynamics, including ventilation and
vertical motion induced by synoptic-scale features such as upper-level troughs. These results
support the idea that synoptic-scale forcing may act as a precursor and driver of RI events.

Similarity, the spatial SST differences between the $P_{80}$ and $P_{20}$ ensembles (Fig. 3a–f)
reinforce the conclusion that thermodynamic conditions alone do not explain the
contrasting intensification outcomes. While some localized differences exceeding $\pm 5°C$
appear at specific time steps, these do not persist or align consistently with the RI period.
The warm anomalies observed in the $P_{20}$ ensemble are mainly displaced to the north and
northeast of Lidia's core. This spatial misalignment suggests that, despite slightly warmer
SSTs, the coupling between oceanic energy supply and inner-core dynamics was likely
suboptimal.



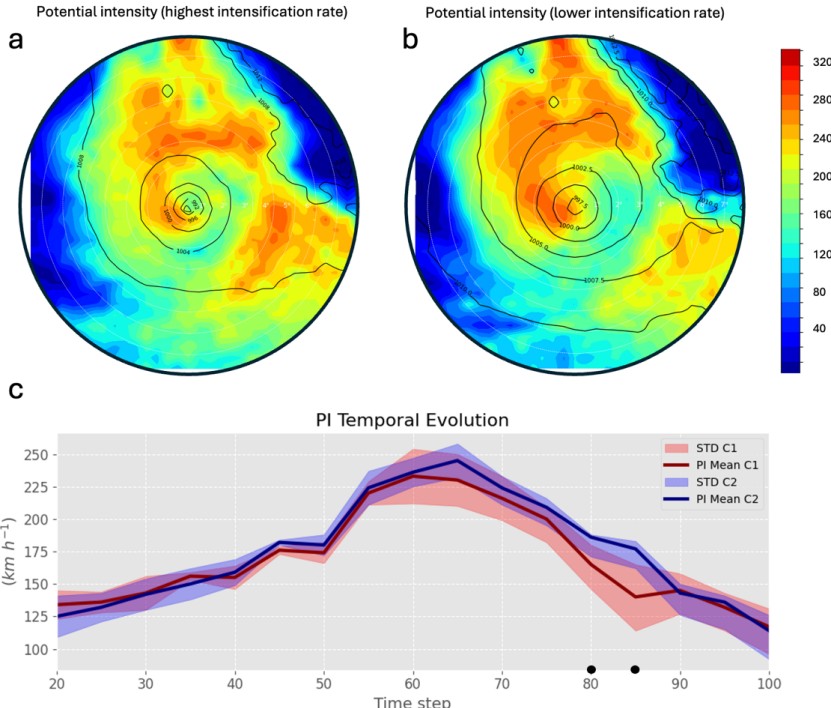

Figure 2. (a) Highest intensification rate PI SCC and (b) lower intensification rate PI SCC
($km\ h^{-1}$) at +55-h and (c) PI calculated within a radial range of with the red (blue) line
representing the higher (lower) IRG. The red (blue) shaded regions indicate the STD for the
highest (lower) IRG.
Statistical significance markers confirm that most SST anomalies are not spatially
coherent enough to produce systematic differences in PI. This is consistent with the similar
PI fields seen in both ensembles (Figs. 2a, b) and the absence of a clear thermodynamic
advantage during the intensification period. Therefore, these SST patterns likely played a
secondary role compared to the dynamically driven processes, such as enhanced vorticity
advection and upper-level divergence.
This supports the notion that SSTs, in this case, provided a necessary but not sufficient
condition for RI. The findings from Bister and Emanuel (2002) and Fischer et al. (2019)
reinforces this view by emphasizing that, without favorable upper-level forcing and
adequate storm structure, warm SSTs alone are insufficient to trigger RI, even when PI
values appear theoretically consistent.

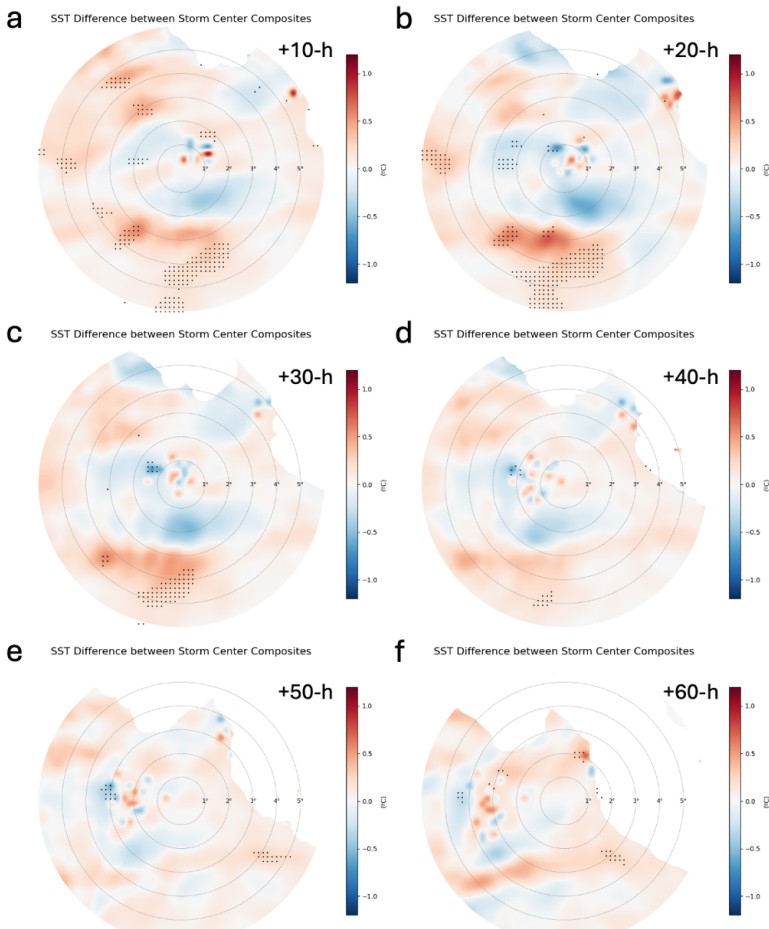

Figure 3. (a-f) SST difference maps ($P_{80}$-$P_{20}$ ensembles; °C) for selected time steps from +10
h to +60 h. Dots indicate regions where differences are statistically significant at the 95%
confidence level.
3.2 Trough interaction and TC rapid intensification
Since thermodynamic factors fail to explain the differences observed among the
ensemble members in Lidia's intensification, we examine the mid- and upper-level dynamic
environment. Figure 4 shows the eastward progression of a trough in both ensemble groups.
The trough is notably broader in the $P_{80}$-ensemble, particularly from time step +55-h. At
250 and 300 hPa (Figs. 4a, b) the isohypses in the $P_{80}$-ensemble exhibit substantial
deformation toward Lidia. The trough deepens further in the $P_{80}$ group at 500 hPa (Fig. 4c),
extending southward to approximately 20°N. This deep mid-tropospheric penetration is
critical because it aligns the trough with the steering and ventilation layers of the tropical
cyclone. The proximity of the trough to Lidia at this level likely contributed to a moist and





unstable environment ahead of the cyclone, while simultaneously promoting vorticity
advection and synoptic-scale positive vertical motions. This configuration aligns with
previous findings on optimal trough-tropical cyclone interactions (e.g., Hanley et al., 2001;
Fischer et al., 2019), which indicate that intensification is favored when the trough
approaches from the northwest at an appropriate distance.
The EFC is computed to diagnose the trough-TC interaction in Hurricane Lidia. The
results show significantly higher EFC values in $P_{80}$- than $P_{20}$-ensemble group. These
significant results are mainly localized between +50 h and +80 h from Lidia RI period (Fig. 5).
$P_{80}$-ensemble is closely aligns with ERA5 reanalysis (exceeding $10\ m\ s^{-1} day^{-1}$ during RI
period). These elevated EFC values are consistent with the findings of DeMaria et al. (1993)
for the North Atlantic basin, where EFC values greater than $10\ m\ s^{-1} day^{-1}$ serve as an
indicator of a trough-TC interaction. Therefore, the obtained EFC values highlight a strong
interaction between Lidia and the trough, suggesting that dynamic forcing, via quasi-
geostrophic approach, enhances vertical motion and upper-level ventilation, potentially
triggering RI. This behavior in the Pacific is analogous to the quasi-stationary effect of the
tropical upper tropospheric trough (TUTT) in the Caribbean, previously analyzed by Sanders
(1975). However, unlike the Caribbean TUTT, which tends to be more persistent and
conducive to cyclogenesis, the trough interacting with Hurricane Lidia in the Pacific is
transient and engages with an already mature TC. This suggests that it plays a crucial role in
Lidia's RI and its subsequent landward turn. Such interactions significantly increase the
potential risk for densely populated areas in Mexico, particularly during the late summer
months, when TCs are most frequent in the eastern Pacific basin (López-Reyes and
Meulenert, 2021).

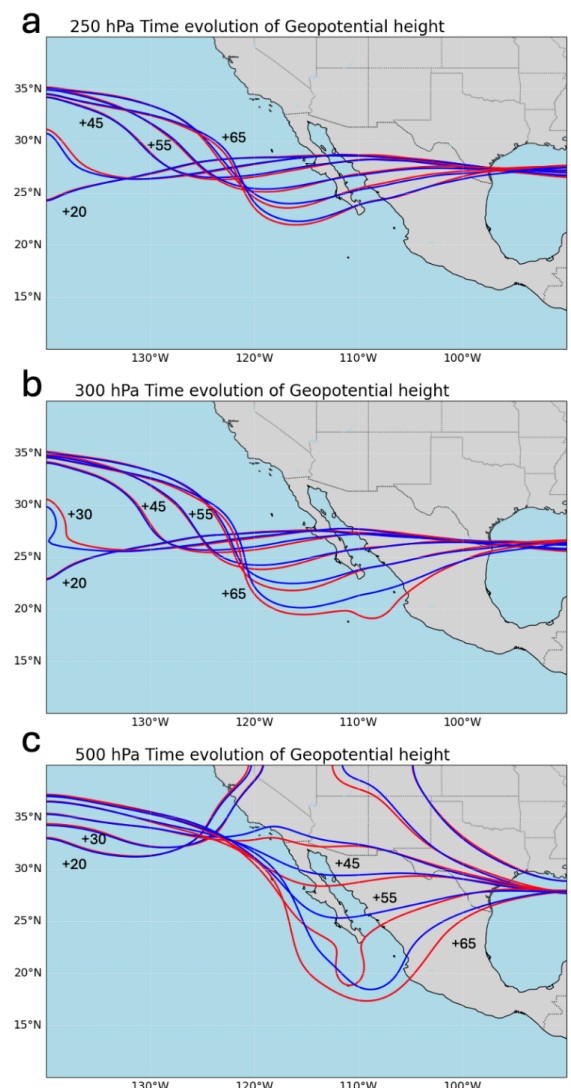

Figure 4. *Z* Composite corresponding to the highest (red contours) and lower (blue contours)
IRG, at (a) 250 hPa, (b) 300 hPa and (c) 500 hPa, before and during the trough-TC interaction.
To assess the dynamical processes supporting Lidia's intensification, EPS outputs during the
RI period are compared with ERA5 fields. Although typically applied in extratropical
contexts, this approach is particularly relevant in the northeastern Pacific, where the
subtropical jet can interact with TCs during autumn, and high-resolution forecast remains
limited.



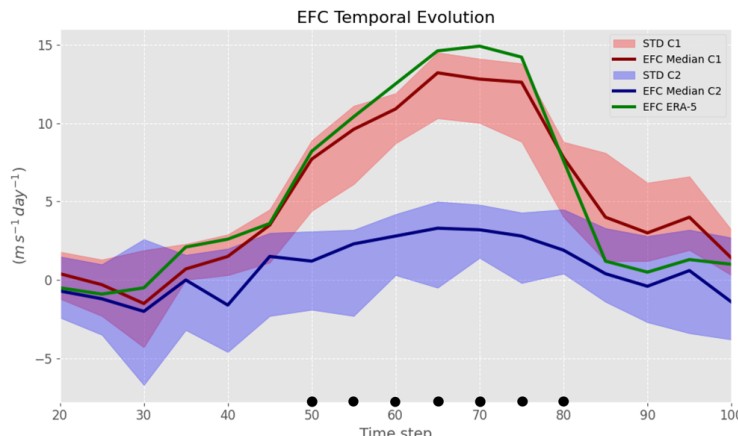

Figure 5. EFC temporal evolution calculated within a radial range of 300 to 600 km at 250 hPa, with the red (blue) line representing the higher (lower) IRG. The red (blue) shaded regions indicate the STD for the highest (lower) IRG, green line represents the EFC based on ERA5 data and dots indicate statistical significance.

In Figures 6a and 6b, the $\nabla \cdot \vec{V}_{ag} > 0$ values, associated with the trough and jet streak, are located to the northeast of Lidia. This configuration strongly favors enhanced upper-level divergence over Lidia and acts as a mechanism that drives upward motions. The quasi-geostrophic $\nabla \cdot \vec{V}_{ag}$ is notably higher in $P_{80}$ than in $P_{20}$-ensemble (Fig. 6a-c); $P_{80}$ closely matches ERA5 across nearly all regions surrounding Lidia (Fig. 6d), suggesting a stronger forcing induced by the interaction with the trough and jet streak.



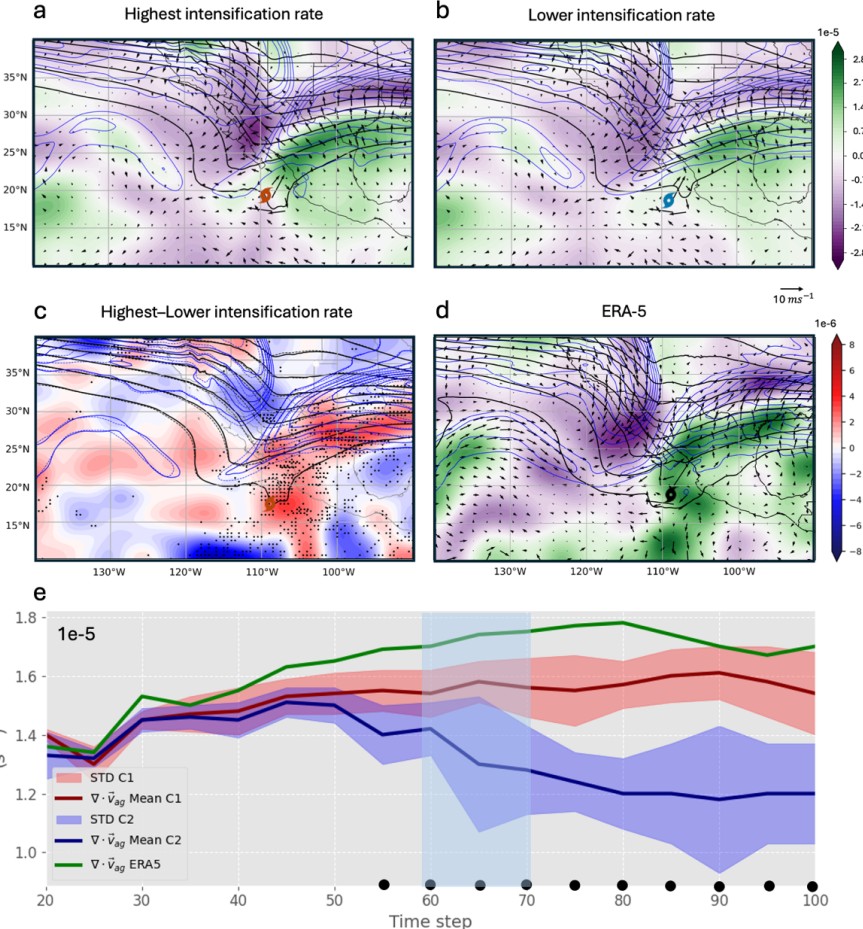


Figure 6. $\nabla \cdot \vec{V}_{ag}$-Composite (shaded; $s^{-1}$), jet stream (blue contours at 10 $ms^{-1}$ intervals)
and $Z$ at 250 hPa (black contours at 20 m intervals) of (a) $P_{80}$, (b) $P_{20}$ IRG (c) $P_{80} - P_{20}$ of
$\nabla \cdot \vec{V}_{ag}$ (shaded; dots indicated statistical significance), solid (dashed) contour represent $Z$
of $P_{80}$ ($P_{20}$) IRG, and (d) same for ERA5 data. e) $\nabla \cdot \vec{V}_{ag}$ Temporal evolution calculated within
a radial range of 500 km at 250 hPa, with the red (blue) line representing the higher (lower)
IRG. The red (blue) shaded regions indicate the STD for the highest (lower) IRG and dots
indicate statistical significance.

Figure 6e reveals distinct differences in the evolution of ageostrophic divergence
between the two ensemble groups. The $P_{80}$ group shows consistently higher values of
ageostrophic divergence, particularly between +50 and +75 h, coinciding with Lidia's RI
period. In contrast, the $P_{20}$ group exhibits lower and declining values during this period,
indicating weaker dynamical forcing. ERA5 closely follows the $P_{80}$ pattern, supporting the
physical credibility of the ensemble signal. These results highlight the role of upper-level
divergence and jet-induced ascent in supporting RI in the $P_{80}$ ensemble.






According to the quasi-geostrophic theory, regions with positive (negative) vorticity advection are associated with upward (downward) vertical motions (Bluestein, 1992). In Figures 7a and 7b, $\vec{V} \cdot \nabla(\vec{\xi} + f)$ is associated with a trough configuration, depicting predominant positive (negative) values in front (behind) of the trough axis. In the same way, $\vec{V} \cdot \nabla(\vec{\xi} + f)$ shows stronger and statistically significant positive values near Lidia's position in $P_{80}$ compared to $P_{20}$-ensemble (Fig. 7c); in addition, a branch with positive vorticity advection values around Lidia is only identified in $P_{80}$-ensemble, and similar to ERA5 (Fig. 7d). The above is consistent with the greater proximity of the trough to Lidia in $P_{80}$-ensemble, highlighting a more intense cyclonic vorticity advection over Lidia (also at earlier time steps; not shown). Therefore, the trough-TC interaction is more robust in $P_{80}$ than in $P_{20}$ as indicated earlier with the EFC metric. This finding shows that a mid-level trough can facilitate the development of a moist layer (Wu et al., 2015), contributing to Lidia intensification.


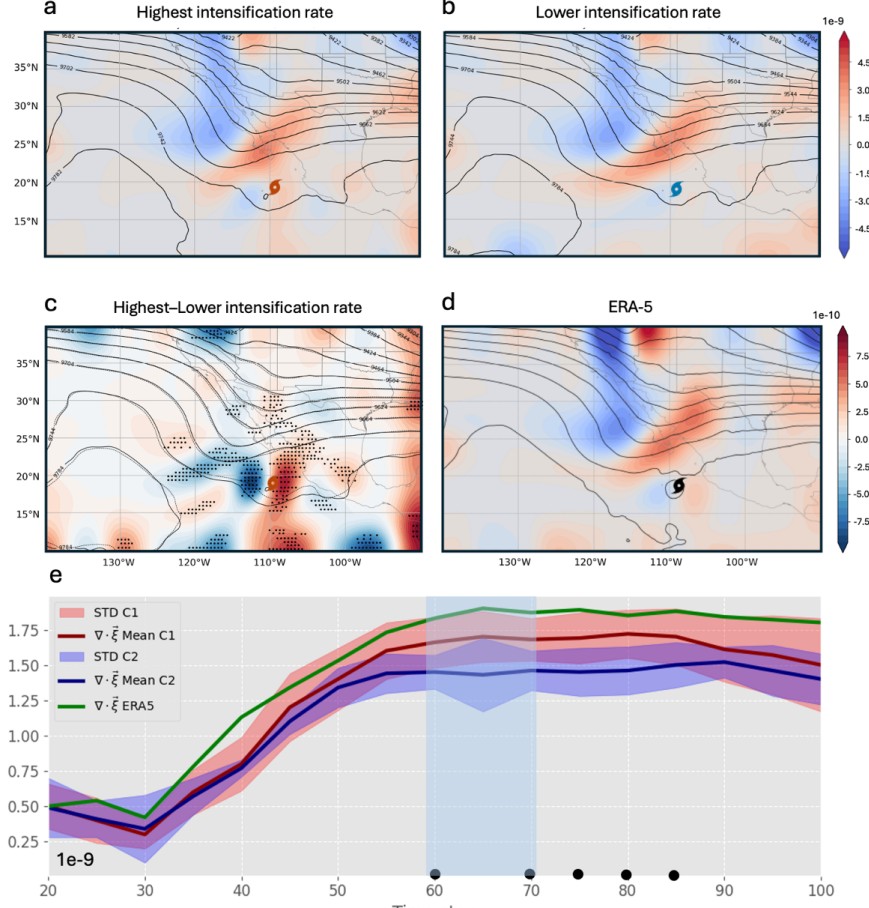






Figure 7. $\vec{V} \cdot \nabla(\vec{\xi} + f)$-Composite (shaded; $s^{-2}$) and $Z$ at 300 hPa (black contours at 20 m
intervals) of (a) the $P_{80}$ (b) $P_{20}$ IRG, (c) $P_{80} - P_{20}$ (shaded; dots indicated statistical
significance), solid (dashed) contour represent $Z$ of $P_{80}$ ($P_{20}$) IRG and (d) same for ERA5 data.
e) $\vec{V} \cdot \nabla(\vec{\xi} + f)$ Temporal evolution calculated within a radial range of 500 km at 500 hPa,
with the red (blue) line representing the higher (lower) IRG. The red (blue) shaded regions
indicate the STD for the highest (lower) IRG and dots indicate statistical significance.

Figure 7e confirms the stronger vorticity forcing in the $P_{80}$-ensemble throughout Lidia's
intensification period. From time step +40 h onward, the $P_{80}$ group exhibits consistently
higher values of vorticity advection, peaking near the RI window (+55 to +70 h), while the
$P_{20}$ group remains consistently weaker, with little variability. The ERA5 line again follows the
$P_{80}$ trajectory, supporting the robustness of the dynamical signal. The statistically significant
differences suggest that enhanced cyclonic vorticity advection, likely associated with the
trough's mid-level deformation, played a crucial role in promoting upward motion and
intensification in the $P_{80}$-ensemble.

The $Q$ field in the $P_{80}$-ensemble (Fig. 8a) shows a more intense upward forcing in the
right region of the trough and extending to the divergence zone at the right entrance of the
jet streak in comparison to $P_{20}$ $Q$ values (Fig. 8b). This contrast becomes even more evident
when considering only RI members within $P_{80}$-ensemble(Figs. 9) are selected and reinforces
the idea of the influence of the trough in Lidia's RI. Based on Eq. (3), negative values of the
forcing term $Q$ correspond to regions of upward vertical motion induced by vorticity
advection via the thermal wind (Dostalek, 2012). The areas surrounding Lidia are strongly
influenced by the dynamical forcing induced by the trough and the jet streak in the $P_{80}$-
ensemble (Fig. 8c). This result is further supported by the ERA5 reanalysis data (Fig. 8d),
which reveals a $Q$ pattern similar to that observed in the $P_{80}$-ensemble, but with greater
intensity (note that ERA5 is only a member, not a composite group). In the absence of
substantial thermodynamic differences (Figs. 2 and 3), these results highlight the dominant
role of dynamic interaction between the trough, the jet streak, and the cyclone during RI.
These findings are particularly relevant for operational forecasting, also demonstrating the
capability  of the ECMWF EPS to simulate Lidia's RI, even under complex extratropical
interactions influences.

The temporal evolution of the Trenberth forcing (Fig. 8e) reveals a clear and consistent
signal in the $P_{80}$-ensemble, with significantly more negative values, indicative of stronger
synoptic-scale upward motion. This enhanced forcing begins well before the onset of Lidia's
RI, peaking between +50 and +70 h. This temporal analysis supports a causal interpretation:
the dynamical forcing precedes and facilitates the RI process, rather than being a
consequence of it. In contrast, the $P_{20}$ group shows much weaker and less coherent values
throughout, indicating and absence of favorable dynamical support for RI.



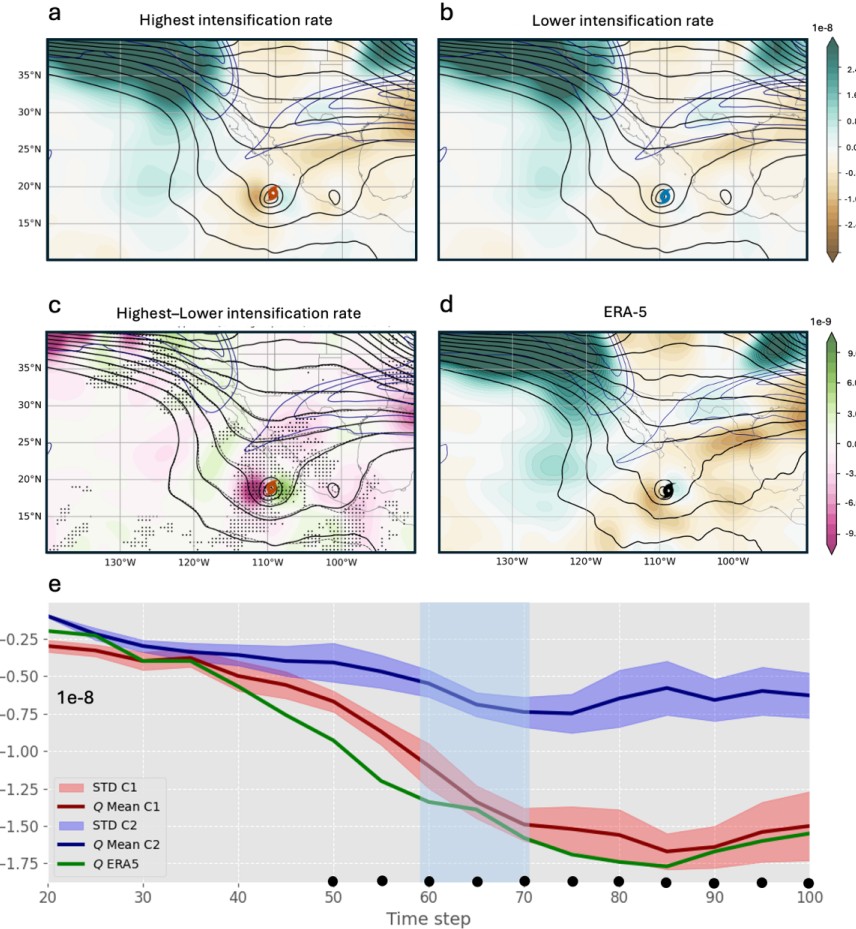

Figure 8. $Q$-Composite (shaded; $Pa \cdot s^{-1}$) at 500 hPa, jet stream (blue contours at $10\ m\ s^{-1}$
intervals) at 250 hPa and $Z$ at 500 hPa (black contours at 20 m intervals) of (a) the $P_{80}$ (b)
$P_{20}$ IRG, (c) $P_{80} - P_{20}$ of $Q$ (shaded; dots indicated statistical significance), solid (dashed)
contour represent $Z$ of $P_{80}$ ($P_{20}$) IRG and (d) same for ERA5 data. (e) $Q$ Temporal evolution
calculated within a radial range of 500 km at 500 hPa, with the red (blue) line representing
the higher (lower) IRG. The red (blue) shaded regions indicate the STD for the highest (lower)
IRG and dots indicate statistical significance.

In $P_{80}$-ensemble, the trough is broader (≈300 km) and positioned closer to Lidia (around
500 km; Figs. 3a, b), in agreement with previous studies showing that favorable trough–TC
interactions occur when the trough lies to the northwest at an optimal distance (Hanley et
al., 2001). Significant differences are observed in both the amplitude and distance relative
to Lidia (Fig. 2c). A similar pattern has been noted in some Atlantic basin cases (Hanley et
al., 2001; Fischer et al., 2019; Sato et al., 2020), where effective trough–TC interactions are
facilitated by a favorable distance, typically between 500-1000 km. Fischer et al. (2019) also
found in a climatological study that TCs in the North Atlantic tend to intensify more rapidly



when the trough is positioned to the northwest, more closely resembling the trough–Lidia pattern in the RI group (Fig. 3a) than in the non-RI group in this work (Fig. 3b).

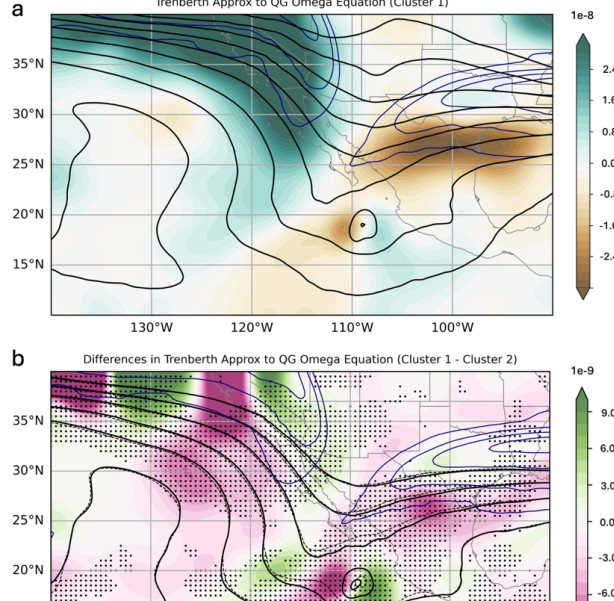

Figure 9. (a) Composite of Trenberth forcing (shaded; $Pa \cdot s^{-1}$) at 500 hPa, jet stream (blue contours at $10\ m\ s^{-1}$ intervals) at 250 hPa and geopotential height at 500 hPa (black contours at 20 m intervals) of RI members, (b) Trenberth forcing (shaded) differences ($P_{80}$-RI members minus $P_{20}$; dots indicated statistical significance), solid (dashed) contour represent geopotential height of RI group.

These findings further support the hypothesis that dynamical forcing triggered Lidia's RI. In the $P_{80}$-ensemble the trough is broader ($\sim 300$ km) and closer to Lidia ($\sim 500$ km; Figs. 10a, b). Evident differences are observed in both the amplitude and distance relative to Lidia (Fig. 10c). A similar configuration has been noted in some Atlantic basin cases (Hanley et al., 2001; Fischer et al., 2019; Sato et al., 2020), where effective trough–TC interactions require a favorable distance. This configuration is associated with different behavior of $\vec{V}_{irr}$ at upper-levels (Figs. 10d-f), where the proximity of the trough's divergence zone enhances ventilation in the $P_{80}$-ensemble (Fig. 10d) with significant $\vec{V}_{irr}$ differences reaching $\sim 4\ m s^{-1}$ to the west-northwest of Lidia. Consequently, the superposition of both divergence zones amplifies the upper-level anticyclonic circulation, consistent with increasing EFC values in $P_{80}$ toward Lidia (Fig. 5) strengthens upward motion and enabling RI.



On the other hand, the VWS remains moderate around Lidia's center in both ensemble
groups, with values between 10-15 $ms^{-1}$ during RI period (Fig. 10g, h). Slightly higher VWS
values are observed to the south of Lidia. To the west and near of Lidia center, VWS values
are higher in $P_{80}$-ensemble (around $\sim 5\ ms^{-1}$; Fig. 10i), though still within favorable ranges
for intensification (Sharma and Varma, 2022). In contrast, regions beyond 2° radial distance
in $P_{80}$-ensemble show significantly lower VWS values, consistent with the position and
shape of the jet stream. In $P_{20}$-ensemble, a stronger jet stream is present north of Lidia,
resulting in a more significant increase in VWS compared to $P_{80}$-ensemble. Thus, the
position and intensity of the jet streak relative to Lidia's position could potentially limit its
intensification in $P_{20}$-ensemble.

The results suggest that the upward motions induced by dynamical mechanisms
associated with Lidia's interaction with a trough are consistent with the greater RH in $P_{80}$,
particularly near the center of Lidia and in the southern region where the trough appears to
enhance its influence (Figs. 10j–l). This region coincides with the trough-cyclone interaction,
where vertical motions are strongly driven by dynamical forcing. The analyzed atmospheric
patterns, including the dynamical forcing associated with the trough and jet streak, suggest
that higher RH in $P_{80}$ may be linked to increased condensation rates during air ascents
around center of Hurricane Lidia, leading to core warming (Emanuel, 1986; Zhang et al.,
2013; Zhang and Emanuel, 2016). This scenario could also decrease VWS, further promoting
Lidia's RI. These findings align with previous studies (Qiu et al., 2020) indicating that TC
intensification can occur even under moderate to high VWS provided that the surrounding
layer remains sufficiently moist.



Figure 10. SCC for the time step 55-h of: $\theta$ $(K)$ at 1.5 PVU for (a) $P_{80}$, (b) $P_{20}$ IRG, and (c) $P_{80} - P_{20}$ IRG, red contours are MSPL; $\left|\vec{V}_{irr}\right|$ $(m\,s^{-1})$, at 200 hPa for (d) $P_{80}$, (e) $P_{20}$ IRG, and (f) $P_{80} - P_{20}$ IRG; $VWS$ $(m\,s^{-1})$ for (g) $P_{80}$, (h) $P_{20}$ IRG and (i) $P_{80} - P_{20}$ IRG, and $RH$ (%) (j) $P_{80}$, (k) $P_{20}$ IRG and (l) $P_{80} - P_{20}$ IRG at 500 hPa.

While previous studies (Braun and Tao, 2000; Rios-Berríos et al., 2018) have shown that increased mid-level humidity can play a key role in RI. The current results indicate that the



556 RH differences between the $P_{80}$ and $P_{20}$ ensembles are minimal and statistically
557 insignificant. This suggests that RH was not the primary driver of RI in this event. Instead,
558 the $P_{80}$ ensemble is characterized by early and sustained dynamic forcing, particularly the
559 strong negative Trenberth forcing observed before the RI onset, which likely initiated
560 upward motion and enhanced upper-level ventilation near the storm core. This synoptic-
561 scale ascent, coupled with the release of latent heat, contributed to a favorable adjustment
562 of the potential vorticity structure in the upper troposphere, reinforcing the outflow and
563 aiding in the vertical alignment of the vortex. As a result of the PV vertical redistribution, a
564 gradual reduction in VWS is observed in the $P_{80}$ group. This reduction is related to higher
565 Trenberth forcing, supporting a causal sequence in which synoptic-scale forcing
566 preconditions, such as strong convection and vortex alignment, subsequently amplify this
567 favorable state, accelerating the intensification process (Chen and Gopalakrishnan, 2019;
568 Komaromi and Doyle, 2018; Stevenson et al., 2014). Figure 11 confirms this evolution:
569 stronger divergence and PV anomalies emerge after the initial forcing, aligning with the
570 onset of RI. The combined evidence supports the conclusion that in the case of Hurricane
571 Lidia, RI was dynamically triggered by the interaction with the upper-level trough and jet
572 stream, with thermodynamic factors, such as PI, SST and RH, playing a secondary, and
573 permissive, rather than a decisive, role.



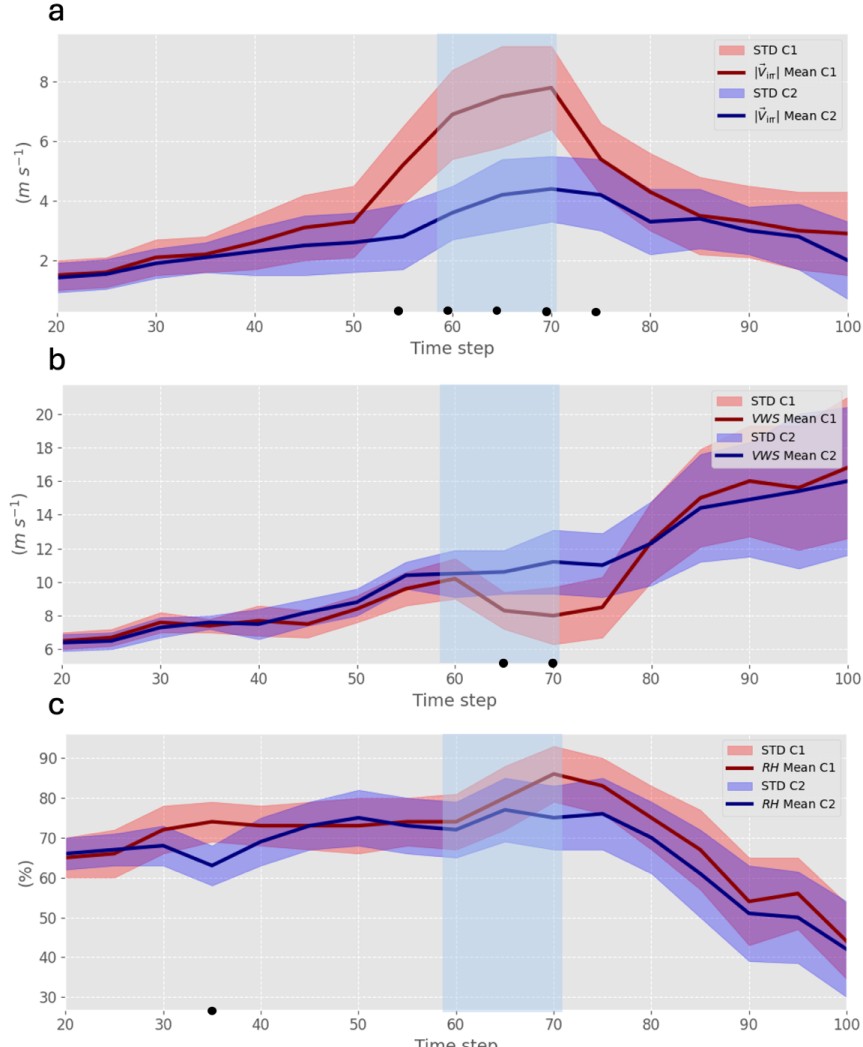

Figure 11. Time evolution of (a) ageostrophic wind at 250 hPa, (b) VWS (850–200 hPa), and
(c) mid-level RH (700–500 hPa) within 500 km of Lidia. Red (blue) lines show the mean for
the higher (lower) IRG; shaded areas indicate STD. Light blue band indicate the RI period.
Black dots denote significant differences.

## 4   Summary and conclusions
This research presents a novel examination of the interaction between a mid- and
upper-level trough and Hurricane Lidia in the northeastern Pacific, a region where studies
are less frequent compared to the Atlantic basin, particularly regarding RI. Since
thermodynamic factors such as PI and SST do not seem to explain the differences observed
between the members of the group in the intensification of Lidia, some dynamic variables



associated with forcings more typical of higher latitudes are analyzed, which usually appear
in the autumn months in subtropical areas of the northeastern Pacific coasts (DiMego et al.,
1976).
Based on previous work in the Atlantic basin (Fischer et al., 2019), which demonstrated
that tropical cyclones experiencing RI often coincide with the presence of an upper-level
trough approaching from the northwest at an optimal distance, our study expands this
framework by demonstrating for the first time a similar dynamic configuration driving RI in
the northeastern Pacific. By analyzing synoptic dynamical indicators, such as the Trenberth
forcing, ageostrophic wind divergence and vorticity advection, we demonstrate how these
dynamical processes play a crucial role in Lidia's RI. The EFC values greater than 10
$ms^{-1}day^{-1}$ in the $P_{80}$-ensemble indicate the trough-TC interaction during the RI period,
reinforcing the critical role of the trough in enhancing vertical motions and upper-level
ventilation. In a context where ocean temperatures are rising and an increasing trend in RI
hurricane frequency  has been documented (Majumdar et al., 2023; Li et al., 2023), this
work provides the first case study of trough-TC interaction leading to RI in the northeastern
Pacific, highlighting the increased proximity and breadth of the trough near Lidia a key driver
of its RI. Unlike Fischer et al. (2019), who focused on climatological composites and
individual case diagnostics in the Atlantic, this study provides a probabilistic ensemble-
based assessment linked to dynamic forcings under realistic forecast uncertainty conditions.
The obtained results underscore the role of dynamical mechanisms, analyzed through
quasi-geostrophic forcing, in triggering significant upward vertical motions that contribute
to the Lidia's RI. These dynamics are evident in the $P_{80}$-ensemble (even more evident in RI
members), where Lidia undergoes RI, showing stronger ageostrophic wind divergence,
enhanced vorticity advection at mid- and upper-levels, and more pronounced Trenberth
forcing, all associated with the influence of the trough. In contrast, in the $P_{20}$-ensemble,
Lidia does not interact with the trough and experiences less favorable conditions. The
proximity and intensity of a jet stream to Lidia's north increase in VWS, which limited the
potential for intensification.
The enhanced Trenberth forcing in the $P_{80}$ ensemble appears several hours before the
onset of RI, indicating that synoptic-scale ascent likely preconditioned the environment
rather than resulting from the intensification itself. This timing supports a causal
interpretation in which the large-scale forcing drives changes within the TC. Following this
initial dynamic trigger, latent heat release near the cyclone core contributed to a favorable
upper-level PV redistribution. This adjustment likely enhanced the upper outflow layer and
contributed to the subsequent reduction in VWS observed in the $P_{80}$ group, further
amplifying the intensification process.
In contrast, relative humidity differences between the two ensembles were small and
not statistically significant, suggesting that moisture availability was not a limiting factor in
this case. While previous studies (Braun & Tao, 2000; Ríos-Berríos et al., 2018) have shown



that enhanced mid-level moisture can favor RI, our results indicate that it played a secondary
role here, acting more as a permissive background condition than an active driver.

By demonstrating the effectiveness of EPS-ECMWF in capturing complex trough-TC
interactions, this study highlights the critical role of EPS as an indispensable tool for
operational forecasting in the northeastern Pacific, especially along the Pacific coast of
Mexico. EPS are particularly valuable for quantifying uncertainty in RI scenarios, which
remain challenging to predict due to the complex dynamical and thermodynamical
processes involved. The present results show that EPS can successfully differentiate
between dynamically favorable and unfavorable environments, even in a context where
high-resolution operational models are not readily available, as is often the case in Mexico.
This makes EPS-based diagnostics especially useful for forecasters operating in data-sparse
or resource-limited settings. In this region, during autumn months, the subtropical jet
stream frequently interacts with TCs, increasing the likelihood of dynamical forcing
mechanisms that can either enhance or inhibit intensification.

This study illustrates how broader and deeper mid-level troughs, such as the one
observed at 500 hPa in Hurricane Lidia, can significantly enhance vertical motion and upper-
level ventilation conducive to RI. Operationally, diagnostic tools such as Trenberth forcing
and EFC metric could be integrated into forecasting to better assess trough-TC interactions.
Measuring these variables in real time would provide forecasters with actionable insights
into the likelihood of RI, particularly when TCs recurve toward the densely populated Pacific
coast of Mexico. Although the limitations of a single case study are evident, we suspect that
other RIs in the northeastern Pacific have been influenced by similar dynamical
mechanisms. However, while our results offer robust evidence from a synoptic-scale
perspective, this study is based on a single case. Future research should expand this
methodology to a broader set of events and explore complementary approaches using
convection-permitting high-resolution simulations. Such simulations would help resolve
inner-core processes and mesoscale interactions that were intentionally simplified in this
study, which focused on evaluating large-scale dynamical forcings. In this regard, the
framework proposed here serves as a cost-effective, scalable strategy to support RI
forecasting in regions with limited access to high-resolution modeling systems and highlights
the continued need to refine multi-scale diagnostic techniques for better understanding and
prediction of TC intensification. Also, expanding this methodology to a broader set of cases
could offer a more comprehensive understanding of trough-TC interactions and their role in
RI, ultimately improving operational forecasting capabilities in this understudied region.

**Declaration of Competing Interest**

The authors declare no conflicts of interest relevant to this study.

**Acknowledgments**



This work was partially supported by the research project PID2023-146344OB-I00
(CONSCIENCE) financed by MICIU/AEI /10.13039/501100011033 and by FEDER, UE, and the
two ECMWF Special Projects (SPESMART and SPESVALE). Mauricio López-Reyes extends his
sincere gratitude to the Institute of IAM of the University of Guadalajara and Instituto
Frontera A.C., for his invaluable support. C. Calvo-Sancho acknowledges the grant awarded
by the Spanish Ministry of Science and Innovation - FPI program (PRE2020-092343).
**Open Research**
The tracking data for Hurricane Lidia can be found in López-Reyes, M. (2024). Atmospheric
data sets can be accessed through the MARS database, hosted by ECMWF, at
https://confluence.ecmwf.int/display/MARS. Additionally, ERA-5 reanalysis data base is
allowed in Climate Data Store (CDS; available at https://climate.copernicus.eu/climate-
reanalysis).
**Author contributions**
Conceptualization: MLR, MLMP, JJGA. Methodology: MLR, MLMP, CCS, JJGA. Project
administration: MLMP. Supervision: MLPM, CCS, JJGZ. Writing-original draft: MLR. Writing-
review and edits: MLR, MLMP, CCS, JJGA.

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
