# Peer review of "Dynamic Forcing Behind Hurricane Lidia's Rapid Intensification"

_EGUsphere, 2025_

## Referee Comment (RC1)

**Review of egusphere-2025-3109 "Dynamic Forcing Behind Hurricane Lidia's Rapid Intensification" By M. Lopez-Reyes, M. L. Martin-Perez, C. Calvo-Sancho, and J. J. Gonzalez-Aleman**

**General comments:**

This study examines an ensemble forecast of Hurricane Lidia (2023), which was a high-impact rapid intensification (RI) event in the eastern North Pacific basin. Lidia interacted with a mid-upper-tropospheric trough near the time of RI. The authors focus on two groups of ensemble members, which were defined as the strongest and weakest 20% of ensemble member forecasts. The authors argue that in the members that intensified the most rapidly, the upper-tropospheric trough provided quasigeostrophic (QG) forcing for ascent over the tropical cyclone (TC), which aided convective development and TC intensification. Alternatively, in a subset of members that did not intensify as rapidly, QG forcing was weaker. The authors argue that because some thermodynamic conditions were similar between the strongest and weakest members, such as sea surface temperatures and potential intensity, it is likely that dynamical forcing played a key role in the RI process.

Overall, I found the manuscript to be well-written and followed a logical order. I believe the authors are aiming to better understand an important physical phenomenon (TC-trough interactions) in a basin that is not well studied. However, I have significant concerns about the authors' arguments over the impact of relative humidity in the RI process. For example, the authors only show the relative humidity field at one forecast time in Fig. 10, which appears to show regions of significant differences between the ensemble groups. However, the authors conclude that "relative humidity differences between the two ensembles were small and not statistically significant". This claim doesn't appear supported by the evidence shown and a more thorough evaluation of the mid-upper-tropospheric moisture evolution is warranted. I also have multiple other scientific comments that I would like the authors to address, which largely pertain to clarifying aspects of the manuscript, including whether or not the TC circulation was filtered in the calculation of the QG forcing for ascent.

**Scientific comments:**

- 1) Line 49: What do the authors mean by ventilation here? Thermodynamic ventilation or ventilating mass?
- 2) Introduction (general comment): The introduction would benefit from a brief elaboration on why Hurricane Lidia is a storm of interest. What were the impacts of the storm?I understand the TC is described in later sections, but at this point, a reader may not know anything about the storm and could be confused over things like when and where RI occurred.
- 3) Lines 121–125: I think one should be careful with language like "main dynamical... drivers of TC RI" when talking about the TC's environment. From a kinematic perspective, TC intensity change is driven by the TC's ability or inability to draw in angular momentum surface via the evacuation of mass out of the boundary layer through

- convection. For example: https://doi.org/10.1002/qj.4133 and https://doi.org/10.1175/MWR-D-24-0029.1
- 4) Line 143: Do the authors mean for "identifying" trough interactions rather than "making"?
- 5) Lines 150–151: While true, should we expect the dynamics of TC–trough interactions to be different in the eastern North Pacific?
- 6) Lines 154–155: I am confused by this sentence. The authors seem to be comparing the Atlantic basin to itself.
- 7) Line 183: I am confused by this claim. Doesn't NOAA routinely run the Hurricane Analysis and Forecast System (HAFS), which is an operational mesoscale model, for TCs in the eastern North Pacific?
- 8) Line 187: What challenges? It would be helpful to be more specific here.
- 9) Line 203: Do the authors mean the forecast spans 96 h?
- 10) Lines 214–215: Does this mean the lowest 20% and top 20% of intensity forecasts? Over what period was this determined? It would be helpful to be more specific.
- 11) Lines 215–216: As a general comment, it would be helpful to explicitly show when RI began in both observations and each model forecast at some point. My apologies if I missed this.
- 12) Line 216: Do the ensemble members have the resolution to truly capture the changes in peak winds? How well did the ensemble members reproduce the best track intensity?
- 13) Lines 223–224: How is shear computed here? Is the TC vortex removed?
- 14) Line 261: What about prior to RI onset? And does this mean for the observed RI period or the simulated RI period in each ensemble member?
- 15) Lines 278–281: It would be helpful to show the evolution of the wind field over this period using the reanalysis.
- 16) Figure 1: The legend of panel a and inset of panel b are quite small and difficult to read.
- 17) Figure 1: While a nice-looking figure, It would be easier to compare the model forecasts with observations in panel b if the two were shown along the same axis. I also recommend revising the x-axis label in panel b from "time step" to "forecast hour" for improved clarity.
- 18) Line 298: "demonstrates" is a strong word choice at this point in the manuscript, considering only one figure has been shown thus far. I recommend revising this word to "suggests".
- 19) Line 305: How was the successful simulation of RI determined? That any forecast point exceeded the 30 kt/24 h threshold?
- 20) Lines 321–322: While I agree with the previous statements, I do not understand how this claim was arrived at. The timing of the synoptic-scale forcing has yet to be shown at this point in the manuscript.
- 21) Lines 325–326: Mostly a curiosity question: what contributes to these large SST differences? Does the model include upwelling processes? Or are these related to differences in TC track?
- 22) Figure 2: Were any aspects of the TC circulation filtered out in the calculation of PI? Also, what are the units for panels a and b? Neither the caption nor figure specifies this.
- 23) Figure 2c caption: What radial range? Please specify in the text.

- 24) Figure 3: Are these TC-centered images? How was the TC center determined in this study?
- 25) Lines 355–357: I question the accuracy of this claim. Yes the authors explore PI and SST, but what tropospheric humidity/ventilation? For example, Fischer et al. (2023; https://doi.org/10.1175/MWR-D-22-0037.1) examined a high-resolution ensemble simulation of a TC–trough interaction case in the North Atlantic. They found a key influence in the timing of RI onset was the degree to which dry air from the nearby trough made it into the TC inner core and eroded convection there (ventilation). Is it not possible humidity/moist entropy differences played an important role here too?
- 26) Lines 362–363: What is the ventilation layer of the TC? I'm not sure what the authors mean by this.
- 27) Line 367: Fischer et al. (2019) found that zonally narrower upper-tropospheric troughs are more favorable for RI. Isn't this the opposite pattern shown in Fig. 4?
- 28) Line 370–371: Both Peirano et al. (2016; https://doi.org/10.1002/2016GL069040) and Fischer et al. (2019) found that EFC tends to be not as important for TC intensification as vertical wind shear. I know it is briefly touched upon later, but showing time series of vertical wind shear magnitude in each ensemble group would be helpful.
- 29) Lines 377–378: What do the authors mean by this phrasing (QG approach)?
- 30) Line 378: Do the authors mean ventilation as in the evacuation of mass? This seems inconsistent with previous literature which refers to ventilation in the thermodynamic sense (e.g., Tang and Emanuel 2010, 2012). I recommend revising to avoid confusion.
- 31) Line 382–383: Is this shown anywhere?
- 32) Lines 394–397: I am confused how the seasonality of the case is important here. Can the authors please clarify?
- 33) Figure 5: Does the uptick in EFC precede the onset of RI? It would be helpful to clarify the time-lag relationship in the text.
- 34) Figure 6e: What is the shaded vertical column between hours 60 and 70? What is this supposed to represent? The figure caption does not specify.
- 35) Lines 431–433: To what extent are the differences near the location of the TC related to the TC circulation itself? Is the TC circulation filtered? If not, the stronger members in the highest intensification rate group may be associated with a vertically-deeper vortex and the advection of the TC's vorticity, rather than the trough's vorticity, at 300 hPa is showing up in this figure. Can the authors please clarify? The wavenumber-1 asymmetry in the QG forcing for ascent near the TC position appears consistent with this.
- 36) Line 438: Aren't the authors looking at upper-levels of the troposphere here rather than "mid-levels"?
- 37) Lines 455–456: Similarly, do the authors mean below the 300 hPa shown in Fig. 7? I wouldn't call 300 hPa mid-levels.
- 38) Lines 469–470: Again, I question whether it's fair to make this claim without examining the role of tropospheric moisture.
- 39) Line 478–479: The differences between the two ensemble groups are not significant, however, until forecast hour 50. According to Fig. 1, this looks to be near the start of RI. Can the authors please clarify?

- 40) Lines 499–502: Actually, Fischer et al. (2019) found TCs in the NW trough cluster have the lowest rate of RI of the three TC–trough configurations examined (see their Fig. 6). RI is preferred when a cutoff low exists to the SW of the TC location.
- 41) Figures 8a-d: Contour labels would be helpful here.
- 42) Figure 8: Again, is the TC vorticity being filtered out here? Otherwise this may just be a representation of differences in TC strength/structure rather than synoptic-scale forcing for ascent.
- 43) Line 493: It is difficult for me to see the broader trough in the P80 ensemble in Fig. 8.
- 44) Lines 525–526: How do the area-averaged shear values compare (e.g., 0–500 km from the TC center)? Such metrics are more commonly used in the literature.
- 45) Lines 534–537: As noted above, how does RH vary prior to the onset of RI? This seems critical to show to properly claim thermodynamic differences did not contribute to the differing intensity evolutions.
- 46) Line 542: Can the authors please clarify how this could decrease VWS?
- 47) Figure 10: These are nice visualizations. However, I do not see any information regarding the spatial scale that these domains span. What is the increment of each radial ring?
- 48) Lines 556–557: What is this claim based on? I see large regions of significant differences in Fig. 10I.
- 49) Line 557: Furthermore, the authors only show RH at one forecast hour. How did RH evolve prior to RI onset?
- 50) Lines 568–573: Hamaguchi and Takayabu (2021; https://doi.org/10.1175/JAS-D-20-0334.1) show how upper-level forcing for ascent can moisten the mid–upper troposphere in tropical depression disturbances. Perhaps a similar sequence is seen here?
- 51) Lines 613–614: Is it fair to say these members have no trough interaction? Or just that the interaction isn't as favorable? There still appears to be a trough in these members.
- 52) Lines 627–628: As noted above, it appears the authors only show one time step. What about at other forecast hours? Furthermore, it appears from Fig. 10I that there are indeed regions of significant differences in RH.

**Typographical errors/suggestions:**

- 1) Line 133: I believe the "Leroux" study is misspelled here.
- 2) Lines 211–212: This seems to be an incomplete sentence.

---

## Author Comment (AC1)

**Response for reviewer 1**

We sincerely thank the reviewer for the thorough and insightful review of our manuscript. We greatly appreciate the constructive comments and suggestions, which have helped us improve the clarity, structure, and scientific rigor of the paper. Many of the points raised were extremely valuable in refining the key arguments and improving the articulation of our results. We have carefully addressed each comment point by point below and revised the manuscript accordingly. We hope the revised version meets the expectations of the reviewer and contributes meaningfully to the understanding of TC rapid intensification processes.

We carefully re-examined the role of RH in the RI process, as this was one of the reviewer's main concerns. We took this comment very seriously and we have performed a detailed re-evaluation of both the figures and the corresponding discussion. Based on this review, we identified inconsistencies between our previous interpretation and the evidence already shown in the results. These issues have now been corrected and clarified. The revised manuscript presents a more balanced and physically consistent explanation of how mid-level RH and dynamical forcing jointly contributed to Lidia's RI. Additional supporting evidence, including an updated description and a new subfigure (Figure 11), has been incorporated to strengthen this section and address the reviewer's concerns fully.

- 1) Line 49: What do the authors mean by ventilation here? Thermodynamic ventilation or ventilating mass?
  - (L51, 120, 150, 195) Thank you for pointing out this ambiguity. In this sentence, we were referring to "ventilating mass,". We have revised the manuscript to clarify this terminology and avoid confusion with thermodynamic ventilation. In fact, throughout the manuscript, when we talk about ventilation, we are referring to divergence air mass. We have made this clearer in the manuscript revisions.
- 2) Introduction (general comment): The introduction would benefit from a brief elaboration on why Hurricane Lidia is a storm of interest. What were the impacts of the storm? I understand the TC is described in later sections, but at this point, a reader may not know anything about the storm and could be confused over things like when and where RI occurred.
  - (L171-176) Thank you for this appropriate suggestion. We agree that adding context to emphasize why Hurricane Lidia is of particular interest enhances the motivation for the study. We have revised the introduction to highlight Lidia's rapid intensification near the Mexican coast, its destructive impacts, and the unexpected nature of the event given the synoptic conditions and forecast challenges. We have added an overview to provide context for Hurricane Lidia and its impacts and importance on lines 161-167.
- 3) Lines 121–125: I think one should be careful with language like "main dynamical... drivers of TC RI" when talking about the TC's environment. From a kinematic perspective, TC intensity change is driven by the TC's ability or inability to draw in angular momentum surface via the evacuation of mass out of the boundary layer through convection. For example: https://doi.org/10.1002/qj.4133 and https://doi.org/10.1175/MWR-D-24-0029.1

(L129-134) Thank you for this valuable clarification and for pointing out the relevant references. We agree that the term "main dynamical drivers of TC RI" may have been misleading in the context of environmental forcing. We have modified the sentence to avoid implying a direct causal role of the environment in TC intensification and instead emphasize its role as a contributing or facilitating factor. The suggested references have also been reviewed and cited accordingly to better contextualize the interpretation within the framework of angular momentum budgets and convective processes.

- 4) Line 143: Do the authors mean for "identifying" trough interactions rather than "making"?
  - (L150) We agree that "identifying" is a more appropriate and precise term in this context.
- 5) Lines 150–151: While true, should we expect the dynamics of TC–trough interactions to be different in the eastern North Pacific?

(L186-191) We thank the reviewer for this important observation. We agree that the fundamental dynamics of TC-trough interactions, such as upper-level divergence, baroclinic forcing, and jet-vortex coupling, are likely governed by similar physical principles across basins. However, we have added a sentence to clarify that what may differ in the eastern North Pacific is the frequency, synoptic configuration, and seasonal context of these interactions, particularly due to the distinct climatology and orography along the Mexican coast. The revised sentence in the manuscript now reads:

"While the underlying dynamics of these interactions may be broadly similar across ocean basins, the eastern North Pacific exhibits unique characteristics that justify a focused investigation. In particular, the variability of the subtropical jet, often modulated by ENSO (Reference), differs from typical Atlantic configurations (Reference), and the recurving behavior of TCs toward the Mexican coastline during late-season months creates a distinct synoptic context"

- 6) Lines 154–155: I am confused by this sentence. The authors seem to be comparing the Atlantic basin to itself.
  - (L186-191) We agree that the original phrasing was unclear. Our intention was to contrast the focus of previous studies, mainly conducted in the Atlantic basin and centered on thermodynamic factors, with our study, which emphasizes dynamic forcing mechanisms in the eastern North Pacific.
- 7) Line 183: I am confused by this claim. Doesn't NOAA routinely run the Hurricane Analysis and Forecast System (HAFS), which is an operational mesoscale model, for TCs in the eastern North Pacific?
  - (L202) We appreciate the reviewer's observation. Indeed, NOAA's Hurricane Analysis and Forecast System (HAFS) has become operational and is routinely run for tropical cyclones in the eastern North Pacific. We have updated the sentence in the manuscript to better reflect this advancement and clarify our intended meaning, which refers to the limited availability of local high-resolution forecasts and analysis products for real-time use in Mexico. The revised sentence now reads:

"While operational mesoscale models such as Hurricane Analysis and Forecast System (HAFS) now provide high-resolution forecasts for tropical cyclones in the eastern North

Pacific, real-time access to their outputs and post-processing capabilities remain limited in Mexico. In contrast, EPS like the European Centre for Medium-Range Weather Forecasts (ECMWF) offer publicly accessible data and have demonstrated strong skill in capturing the uncertainty associated with complex and potentially high-impact TC scenarios. Therefore, ensemble-based diagnostics remain a valuable and scalable approach for assessing TC behavior and RI risks, particularly in resource-constrained forecasting environments such as Mexico"

8) Line 187: What challenges? It would be helpful to be more specific here.

(L205) We have revised the paragraph to specify the operational challenges in Mexico, including limited access to real-time model outputs, lack of post-processing infrastructure, and insufficient human resources.

9) Line 203: Do the authors mean the forecast spans 96 h?

(L226) Yes, thank you for the clarification request. We have revised the sentence to explicitly state that the forecast spans 96 h.

10) Lines 214–215: Does this mean the lowest 20% and top 20% of intensity forecasts? Over what period was this determined? It would be helpful to be more specific.

(L239-243) We clarified that the percentiles were based on the minimum central pressure reached by each ensemble member during the 24-h period of most rapid intensification (Oct 9, 00:00 UTC - Oct 10, 00:00 UTC) and verified using wind speed to meet the RI definition (≥54 km/h in 24 h).

11) Line 216: Do the ensemble members have the resolution to truly capture the changes in peak winds? How well did the ensemble members reproduce the best track intensity?

We acknowledge the limitations in resolution inherent to global ensemble models. While most members underestimated Lidia's peak intensity, some were able to simulate its RI. These members also reproduced a synoptic configuration consistent with a mid-level trough interacting with the cyclone. This suggests that, despite their lower resolution compared to mesoscale models, global ensembles can still capture key signals associated with RI events, particularly under favorable dynamical forcing. Additionally, we have included in the reviewer's material a supplementary figure showing the evolution of maximum wind speed for each ensemble member to illustrate the variability in simulated intensities.

**12) Lines 223–224: How is shear computed here? Is the TC vortex removed?**

In our methodology the effect of vortex is not removed, we computed VWS as the vector difference between the horizontal wind averaged over the 200 hPa level and the horizontal wind averaged over the 850 hPa level, within a radius of 300 km centered on the tropical cyclone.

**13) Line 261: What about prior to RI onset? And does this mean for the observed RI period or the simulated RI period in each ensemble member?**

(L298-300) We appreciate this important clarification. The EFC was computed for the entire 96-hour forecast period starting from the model initialization time, thus encompassing the pre-RI, RI, and post-RI phases. This was applied consistently to both ERA5 and all ensemble members to allow for a comprehensive temporal comparison.

**14) Lines 278–281: It would be helpful to show the evolution of the wind field over this period using the reanalysis.**

We have included the evolution of the wind field from ERA5 reanalysis at 250 hPa, along with the geopotential height anomalies at 250 and 500 hPa. These additions help illustrate the synoptic structure of the jet stream and highlight the anomalous nature of the event. The figures were added in the appendix of the manuscript.

15) Figure 1: The legend of panel a and inset of panel b are quite small and difficult to read.

The font size of the legend in panel (a) and the inset in panel (b) has been increased to improve readability in the revised figure.

16) Figure 1: While a nice-looking figure, It would be easier to compare the model forecasts with observations in panel b if the two were shown along the same axis. I also recommend revising the x-axis label in panel b from "time step" to "forecast hour" for improved clarity. In the same way in all figures.

In the revised Figure 1b, we now plot both the model forecasts and ERA5 observations using the same vertical axis to facilitate direct comparison. Additionally, the x-axis label has been changed from "time step" to "forecast hour" to improve clarity.

17) Line 298: "demonstrates" is a strong word choice at this point in the manuscript, considering only one figure has been shown thus far. I recommend revising this word to "suggests".

| of the manuscript. |  |  |  |
|--------------------|--|--|--|
|                    |  |  |  |
|                    |  |  |  |
|                    |  |  |  |
|                    |  |  |  |
|                    |  |  |  |
|                    |  |  |  |
|                    |  |  |  |
|                    |  |  |  |
|                    |  |  |  |
|                    |  |  |  |
|                    |  |  |  |
|                    |  |  |  |
|                    |  |  |  |
|                    |  |  |  |
|                    |  |  |  |
|                    |  |  |  |
|                    |  |  |  |
|                    |  |  |  |
|                    |  |  |  |
|                    |  |  |  |
|                    |  |  |  |
|                    |  |  |  |

18) Line 305: How was the successful simulation of RI determined? That any forecast point exceeded the 30 kt/24 h threshold?

(L241-243) We now clarify that a successful simulation of RI was defined as any ensemble member exceeding the 30 kt / 24 h intensification threshold during any 24-hour window within the forecast period. The criteria were added explicitly in the revised text.

19) Lines 321–322: While I agree with the previous statements, I do not understand how this claim was arrived at. The timing of the synoptic-scale forcing has yet to be shown at this point in the manuscript.

(L358-362) Thank you for this important observation. We agree that the timing of the synoptic-scale forcing had not been explicitly shown at that point. To clarify this, we have added a transitional sentence at the end of the paragraph that introduces the upcoming analysis of the timing and structure of synoptic forcing during RI. This helps improve the logical flow and prepares the reader for the figures and diagnostics that follow.

20) Lines 325–326: Mostly a curiosity question: what contributes to these large SST differences? Does the model include upwelling processes? Or are these related to differences in TC track?

Since the operational IFS model was used in this study, we believe that upwelling processes are explicitly represented, as the IFS couples the ocean and the atmosphere through a fully interactive surface scheme. Therefore, the large SST differences are likely influenced by both local upwelling induced by the cyclone's circulation and variations in TC track among ensemble members.

ECMWF. (2018). IFS upgrade brings more seamless coupled forecasts. ECMWF Newsletter, 156, 1–5. <a href="https://www.ecmwf.int/sites/default/files/elibrary/2018/18873-ifsupgrade-brings-more-seamless-coupled-forecasts.pdf">https://www.ecmwf.int/sites/default/files/elibrary/2018/18873-ifsupgrade-brings-more-seamless-coupled-forecasts.pdf</a>

21) Figure 2: Were any aspects of the TC circulation filtered out in the calculation of PI? Also, what are the units for panels a and b? Neither the caption nor figure specifies this.

Thank you for this useful observation. We confirm that no filtering was applied to remove TC circulation in the calculation of PI. The calculation follows the standard methodology (following to Bister and Emanuel (1998) and Gilford (2021)), using environmental values from IFS. We have clarified this point in the revised figure caption. In addition, we have added the units for panels (a) and (b) in the caption to ensure clarity.

22) Figure 2c caption: What radial range? Please specify in the text.

We have clarified in the figure caption that the PI shown in panel 2c was computed over a radial range of 6°, consistent with the methodology described earlier in the text.

23) Figure 3: Are these TC-centered images? How was the TC center determined in this study?

Yes, Figure 3 shows TC-centered composites. The storm center in each member was

24) Lines 355–357: I question the accuracy of this claim. Yes the authors explore PI and SST, but what tropospheric humidity/ventilation? For example, Fischer et al. (2023; https://doi.org/10.1175/MWR-D-22-0037.1) examined a high-resolution ensemble simulation of a TC-trough interaction case in the North Atlantic. They found a key influence in the timing of RI onset was the degree to which dry air from the nearby trough made it into the TC inner core and eroded convection there (ventilation). Is it not possible humidity/moist entropy differences played an important role here too?

(L401-403 and L407-416) We thank the reviewer for this insightful observation and agree that tropospheric humidity and ventilation effects can play an important role in modulating RI onset, particularly in trough-TC interaction scenarios (as you mentioned Fischer et al., 2023). While our study primarily emphasizes dynamical forcing, we acknowledge that differences in moist entropy and humidity, especially mid-level intrusions from the trough, may have also influenced convection and RI timing. We have added a sentence in the discussion section to reflect this possibility and cite Fischer et al. (2023) accordingly.

25) Lines 362–363: What is the ventilation layer of the TC? I'm not sure what the authors mean by this.

(L401-403) We appreciate the observation and agree that the term "*ventilation layer*" may have been unclear or potentially misleading. In our manuscript, we used this term to refer to the greater southward elongation of the mid- and upper-level trough (notably around 500-250 hPa) observed in the RI ensemble members, which brings the trough anomalously close to the tropical cyclone center. This proximity facilitates interaction with the steering flow and may influence upper-level divergence. However, we recognize that "ventilation layer" is often associated with environmental dry-air intrusion or vertical wind shear, and thus we have revised the text to avoid confusion. Thank you for pointing this out.

26) Line 367: Fischer et al. (2019) found that zonally narrower upper-tropospheric troughs are more favorable for RI. Isn't this the opposite pattern shown in Fig. 4?

(L407-416) Thank you for this important comment. While the RI-associated trough appears broader, Figure 4c (+45 h, +55 h) reveals a more pronounced southward elongation in the RI group. This configuration may enhance the interaction with the TC core, partially offsetting the zonal extent noted in Fischer et al. (2019). A clarification has been included in the revised text.

"Such a configuration is consistent with previous findings on optimal trough-tropical cyclone interactions (e.g., Hanley et al., 2001; Fischer et al., 2019), which suggest that intensification is favored when the trough approaches from the northwest at an appropriate distance. Although Fischer et al. (2019) noted that narrower upper-tropospheric troughs may be more conducive to RI, the enhanced interaction observed here may result from the deeper and more equatorward positioning of the broader trough in the RI group (particularly at +45 h and +55 h in Fig. 4c)"

27) Line 370–371: Both Peirano et al. (2016; https://doi.org/10.1002/2016GL069040) and Fischer et al. (2019) found that EFC tends to be not as important for TC intensification as vertical wind shear. I know it is briefly touched upon later, but showing time series of vertical wind shear magnitude in each ensemble group would be helpful.

An additional discussion on VWS has now been incorporated into the revised manuscript, highlighting its relative contribution compared to eddy flux convergence in this case. Moreover, time series of VWS magnitude for both ensemble groups (P80 and P20) have been added, allowing for a clearer comparison of the temporal evolution of shear prior to and during RI. This addition strengthens the interpretation that reduced VWS in the P80.

28) Lines 377–378: What do the authors mean by this phrasing (QG approach)?

(L426-430) By "QG approach," we refer specifically to the use of quasi-geostrophic theory to estimate vertical motion forcing, particularly via the QG omega equation in Trenberth form. We have revised the text to clarify this point accordingly.

29) Line 378: Do the authors mean ventilation as in the evacuation of mass? This seems inconsistent with previous literature which refers to ventilation in the thermodynamic sense (e.g., Tang and Emanuel 2010, 2012). I recommend revising to avoid confusion.

(L429) We appreciate the reviewer's insightful observation. We acknowledge that our original use of the term ventilation could be misleading, as it may be interpreted in the thermodynamic sense described by Tang and Emanuel (2010, 2012), rather than in the dynamical context we intended. To avoid confusion, we have revised the sentence to refer instead to upper-level divergence, which more accurately reflects the dynamic interaction between the trough and the tropical cyclone within the quasi-geostrophic framework used in our analysis.

**30) Line 382–383: Is this shown anywhere?**

While we do not explicitly show the impact of the trough on Lidia's track in the current version of the manuscript, preliminary analyses from earlier model initializations revealed that the presence or absence of interaction with the trough significantly influenced Lidia's trajectory. Specifically, some members without a well-defined trough steered the cyclone westward, away from Mexico. However, to maintain focus on the RI mechanisms and avoid diverging from the central scope of the study, we decided not to include these trajectory differences in the final text. We have revised the sentence accordingly to prevent unsupported implications.

31) Lines 394–397: I am confused how the seasonality of the case is important here. Can the authors please clarify?

(L446-449) We appreciate the reviewer's request for clarification. We have now explained in the revised text that the seasonality of this case is relevant because, during boreal autumn, tropical cyclones in the eastern North Pacific tend to recurve more frequently due to the strengthening of the midlatitude westerlies. This increases the likelihood of interactions with upper-level troughs or the subtropical jet, which can significantly influence both the intensity and trajectory of tropical cyclones approaching the Mexican coastline.

32) Figure 5: Does the uptick in EFC precede the onset of RI? It would be helpful to clarify the time-lag relationship in the text.

We appreciate the reviewer's suggestion to clarify the temporal relationship between EFC and the onset of RI. We have added a sentence to the revised manuscript to address this. Specifically, we note that differences in EFC between the P80 and P20 ensembles begin to emerge several hours prior to RI onset (from +40 h), becoming statistically significant at +50 h. These differences are maintained throughout the RI period, supporting the interpretation that enhanced EFC may have contributed to the initiation and maintenance of RI.

33) Figure 6e: What is the shaded vertical column between hours 60 and 70? What is this supposed to represent? The figure caption does not specify.

The shaded vertical column between forecast hours +55 and +70 in Fig. 6e represents the period during which RI occurred in Hurricane Lidia, based on the official best-track data. We have now clarified this explicitly in the figure caption.

34) Lines 431–433: To what extent are the differences near the location of the TC related to the TC circulation itself? Is the TC circulation filtered? If not, the stronger members in the highest intensification rate group may be associated with a vertically-deeper vortex and the advection of the TC's vorticity, rather than the trough's vorticity, at 300 hPa is showing up in this figure. Can the authors please clarify? The wavenumber-1 asymmetry in the QG forcing for ascent near the TC position appears consistent with this.

(L491-494) We thank the reviewer for this insightful observation. In the original analysis, a Gaussian spatial filter with  $\sigma$  = 1.5° was applied uniformly to all fields to remove small-scale noise while retaining the synoptic-scale structure. However, we agree that the TC's own cyclonic vorticity could still contribute to the local advection patterns near the storm center.

To further minimize this contamination, we have included additional figures in the Appendix where the Gaussian filter is applied locally to the TC region only, within an 800 km radius around the TC center, with a continuous cosine taper toward the surrounding domain to avoid sharp spatial transitions.

The results confirm that, after localized filtering, a smoother synoptic-scale forcing pattern emerges: the QG ascent forcing remains evident but is now more clearly associated with the upper-level trough and jet-streak interaction rather than the TC's internal vorticity advection.

We have revised the methodology section to explicitly describe this additional test and added a note in the main text referring readers to the Appendix for the corresponding filtered fields.

"It is worth noting that the QG ascent patterns near the TC center may partially reflect contributions from the TC's own circulation. This implies that some contamination from the TC's inner-core vorticity cannot be completely ruled out. To assess this, we performed an additional localized filtering applied exclusively to the TC circulation, which effectively removes most of the mesoscale contribution of the vortex. As shown in Appendix A3, the resulting Trenberth forcing field reveals a clearer synoptic-scale signal associated with the trough and the jet-streak interaction, supporting that the large-scale forcing dominates despite minor contamination near the TC center."

In addition, we have calculated the storm centered QG forcing fields with a Gaussian filter to identify the synoptic contribution that can intensify the vertical movements (see supplementary Figure S1).

35) Line 438: Aren't the authors looking at upper-levels of the troposphere here rather than "mid-levels"?

(L495-496) We agree that the choice of terminology could be more precise. While the irrotational wind component was analyzed at upper-tropospheric levels (specifically 250 hPa), the vorticity advection and Trenberth QG forcing were computed at what is typically considered mid-tropospheric levels, notably 500 hPa, following conventions described in Bluestein (1992). Therefore, our analysis combines both upper- and mid-tropospheric levels, depending on the specific variable being evaluated. We have clarified this distinction in the revised manuscript to avoid confusion.

36) Lines 455–456: Similarly, do the authors mean below the 300 hPa shown in Fig. 7? I wouldn't call 300 hPa mid-levels.

(L513) To avoid ambiguity, we have revised the manuscript to explicitly.

37) Lines 469–470: Again, I question whether it's fair to make this claim without examining the role of tropospheric moisture.

(L524-531 and L536-538) We agree that tropospheric moisture plays a key role in tropical cyclone intensification. While our current analysis focused primarily on dynamical factors such as the mid-to-upper-level trough interaction and associated forcing, we acknowledge that differences in tropospheric humidity, especially mid-level dry air intrusions, could have influenced the timing and magnitude of RI in the ensemble. As such, we have added a clarifying sentence in the revised manuscript to acknowledge this limitation and the potential role of moisture, citing recent work (e.g., Fischer et al. 2023) that underscores this influence.

38) Line 478–479: The differences between the two ensemble groups are not significant, however, until forecast hour 50. According to Fig. 1, this looks to be near the start of RI. Can the authors please clarify?

(L545) We have clarified in the manuscript that the statistically significant differences in Trenberth forcing between the two ensemble groups begin around +50 h, which is slightly before the onset of Lidia's RI, occurring at +55 h. This temporal sequence supports a causal relationship, in which the enhanced synoptic-scale forcing could help trigger RI, rather than simply reflecting its effects.

39) Lines 499–502: Actually, Fischer et al. (2019) found TCs in the NW trough cluster have the lowest rate of RI of the three TC–trough configurations examined (see their Fig. 6). RI is preferred when a cutoff low exists to the SW of the TC location.

(L569-574) We thank the reviewer for pointing out this inconsistency. Indeed, Fischer et al. (2019) found that TCs embedded in northwestward trough configurations exhibited the lowest RI rates, while the cutoff low to the southwest pattern was more favorable for intensification. We regret the misrepresentation and have corrected this statement in the manuscript to better reflect their findings.

40) Figures 8a-d: Contour labels would be helpful here.

We have added contour labels to Figures 8a-d to improve clarity and readability.

41) Figure 8: Again, is the TC vorticity being filtered out here? Otherwise, this may just be a representation of differences in TC strength/structure rather than synoptic-scale forcing for ascent.

We appreciate the reviewer's insightful observation. Indeed, no explicit filtering of the TC vortex was applied in the Trenberth forcing calculation, which means that part of the signal may reflect differences in the TC's own structure and intensity between ensemble members. However, we note that the vorticity and geopotential height fields used in the QG diagnostic are primarily sensitive to synoptic-scale features at the analyzed resolution, and that the coherent wavenumber-1 pattern observed is consistent with trough-induced asymmetries.

We have nuanced this in the manuscript to have a broader perspective in the discussion and interpretation of the results.

"However, we acknowledge that part of this signal also reflects the contribution from the TC circulation itself. Nonetheless, at the synoptic scale, coherent differences associated with the trough's position and structure are clearly discernible between ensemble groups."

We have also performed the filtering suggested by the reviewer using a storm-centered Gaussian mask, and the results are now included in Appendix A3. These additional fields confirm that the synoptic-scale patterns remain even after removing the TC inner-core circulation, supporting the interpretation presented in the main manuscript.

42) Line 493: It is difficult for me to see the broader trough in the P80 ensemble in Fig. 8.

We acknowledge that the broader structure of the trough is not easily distinguishable in Fig.8, particularly given the complex synoptic environment and overlapping features. However, a slightly larger amplitude of the contours near the TC can still be identified in the P80-ensemble (Fig.8c), which may reflect the subtle but meaningful differences in the trough configuration that favor stronger dynamic forcing.

When we did the calculation, we found that the difference in the width of the trough is around 300 km at the latitude where Lidia is located (differences between continuous and discontinuous contours).

43) Lines 525–526: How do the area-averaged shear values compare (e.g., 0–500 km from the TC center)? Such metrics are more commonly used in the literature.

Following your recommendation, we have now clarified that the vertical wind shear values shown in Figure 9 were area-averaged within a 500 km radius from the TC center, which is consistent with commonly used metrics in previous literature. This detail has been explicitly included in the Methods section and in the caption of Figure 9 to improve clarity and reproducibility.

44) Lines 534–537: As noted above, how does RH vary prior to the onset of RI? This seems critical to show to properly claim thermodynamic differences did not contribute to the differing intensity evolutions.

(L611-614 and L619-624) Following this suggestion, we examined the RH temporal evolution at different forecast hours preceding the onset of RI for both ensemble groups. The analysis confirms that differences between the two groups were already evident prior to the onset of RI, with higher mid-tropospheric RH in the rapidly intensifying (P80) members compared to the weak-intensifying group. To strengthen this point, we have included an additional subpanel in Figure 11, showing the evolution of RH at 500 hPa, which complements the patterns presented in Figure 10j-I. This addition helps provide a consistent depiction of the thermodynamic environment and reinforces our argument that RH differences contributed to the contrasting intensity evolutions.

**45) Line 542: Can the authors please clarify how this could decrease VWS?**

(L618-623)The decrease in VWS mentioned in the manuscript refers to a local reduction in the vertical wind gradient over the cyclone core during the interaction with the upper-level trough. This reduction results from a combination of dynamical and thermodynamic processes. Dynamically, the enhanced upper-level divergence associated with the jet streak and trough interaction promotes mass removal aloft, which modifies the upper-tropospheric flow and leads to weaker winds directly above the storm center. Thermodynamically, the higher mid-tropospheric RH in the P80 ensemble supports stronger and more vertically aligned convection, enhancing latent heat release and intensifying the upper-level outflow. This process reinforces the warm-core structure and induces a local reconfiguration of the wind profile that effectively reduces the vertical shear experienced by the storm (Riemer and Montgomery, 2011; Ge et al., 2013; Tang and Emanuel, 2012; Ryglicki et al., 2019).

To improve clarity, the manuscript has been revised to explicitly state that the reduction in VWS refers to the effective shear acting on the storm's inner core, which decreases as a consequence of both enhanced dynamical forcing and moisture-supported convective alignment.

The citations have been added to the manuscript.

46) Figure 10: These are nice visualizations. However, I do not see any information regarding the spatial scale that these domains span. What is the increment of each radial ring?

We agree that the visualization did not clearly indicate the radial scale, making it difficult to identify the increment of each ring. To address this, we have clarified the figure caption, which now specifies that each concentric ring corresponds to an increment of 1°, and that the outermost circle representing the SCC has a radius of 8°.

47) Lines 556–557: What is this claim based on? I see large regions of significant differences in Fig. 10I.

(L650-658) We thank the reviewer for this accurate observation. We agree that Figure 10l indeed shows statistically significant differences in RH, particularly within and south of the storm core. After re-examining the results, we found that the P80 ensemble exhibits RH values approximately 10% higher than the P20 ensemble in these regions. We have

revised the text to acknowledge this finding and to clarify that these differences are both spatially coherent and statistically significant. This correction strengthens the interpretation that locally enhanced humidity contributed to sustaining deep convection and promoting a more vertically aligned vortex structure in the P80 ensemble (in agree with previous studies, e.g., Alland et al., 2021; Tang and Emanuel 2010). The changes in the manuscript are as follows:

"While the mean RH differences between the P\_80-ensemble and P\_20-ensemble were modest in magnitude, Figure 10l reveals statistically significant anomalies of approximately 10% near the storm center and along its southern flank. These regions of enhanced mid-tropospheric moisture likely played an active role in sustaining deep convection and facilitating the vertical alignment of the vortex, consistent with the stronger and more organized convective structure observed in the P\_80-ensemble. This behavior is in agreement with previous studies (e.g., Alland et al., 2021; Tang and Emanuel, 2010), which demonstrated that higher mid-level humidity reduces ventilation and supports the maintenance of deep, symmetric convection even under moderate vertical wind shear."

The authors believe that these modifications and clarifications regarding the RH results provide more robust physical evidence and lead to a more balanced assessment of the relative contributions of dynamic and thermodynamic forcings.

**48) Line 557: Furthermore, the authors only show RH at one forecast hour. How did RH evolve prior to RI onset?**

(L650-658) In response, we added an additional panel to Figure 11 showing the temporal evolution of RH at 500 hPa. This new analysis reveals that significant differences in RH emerged several hours prior to the onset RI, particularly near the storm core and southern sector. In the previous version, we had computed an average between 700 and 500 hPa, which masked these mid-level differences. The revised figure now highlights the temporal progression of RH and its consistent contrast between the two ensembles, strengthening the evidence for the role of moisture in preconditioning the environment for RI. We sincerely thank the reviewer for pointing out this important detail, which has improved both the clarity and robustness of the results.

49) Lines 568–573: Hamaguchi and Takayabu (2021; https://doi.org/10.1175/JAS-D-20-0334.1) show how upper-level forcing for ascent can moisten the mid–upper troposphere in tropical depression disturbances. Perhaps a similar sequence is seen here?

(L660-667) We carefully reviewed Hamaguchi and Takayabu (2021) and compared their findings with our results. Their study shows that TUTTs can precede deep convection by forcing synoptic-scale ascent that moistens the mid–upper troposphere on the southeast flank of the trough, this dynamical moistening preconditions the environment for subsequent convective amplification. In our case, we find a very similar sequence. Specifically, the negative Trenberth forcing and enhanced upper-level divergence in the P80 ensemble appear several hours prior to RI, followed by a significant increase of 500 hPa RH (≈10%) near the storm center and along the southern flank, and then a reduction of VWS as the vortex becomes better aligned. Thus, although Hamaguchi and Takayabu

(2021) focus on tropical-depression-type disturbances, the physical pathway they document, upper-level dynamical ascent → mid-tropospheric moistening → deeper, more symmetric convection, maps well onto our RI case. We have added text in the manuscript to make this connection explicit and to note that this paper helped us articulate the linkage between the trough-forced ascent, the RH evolution, and the subsequent shear reduction.

50) Lines 613–614: Is it fair to say these members have no trough interaction? Or just that the interaction isn't as favorable? There still appears to be a trough in these members.

(L746-749) We agree with the reviewer's observation. Indeed, a trough is still present in the weakly intensifying members, but the interaction between the trough and the tropical cyclone is less favorable in terms of both position and amplitude. We have revised the corresponding sentence to clarify that the difference lies not in the absence of interaction, but rather in its reduced dynamical effectiveness and weaker coupling with the upper-level flow.

51) Lines 627–628: As noted above, it appears the authors only show one time step. What about at other forecast hours? Furthermore, it appears from Fig. 10l that there are indeed regions of significant differences in RH.

(L761-773) We acknowledge that the initial version showed RH at only one forecast hour, which limited the temporal interpretation. In the revised manuscript, we have included an additional panel in Figure 11 displaying the temporal evolution of RH at 500 hPa prior to the onset of RI. This new analysis reveals that the RH differences between the two ensembles were not transient but developed progressively several hours before intensification after QG forcing. The P80 ensemble consistently exhibits a  $\sim\!10$  % higher RH near the storm center and along its southern flank, as also evident in Figure 10I, where these anomalies are statistically significant. These revisions clarify the temporal behavior of mid-tropospheric humidity and reinforce its role, together with the dynamical forcing, in preconditioning the environment for Lidia's rapid intensification.

**Typographical errors/suggestions:**

1) Line 133: I believe the "Leroux" study is misspelled here.

We have corrected the reference.

2) Lines 211–212: This seems to be an incomplete sentence.

We acknowledge the issue and have revised the sentence to ensure it is complete and grammatically correct.

**Response for reviewer 2**

We sincerely thank for the positive and encouraging evaluation of our manuscript. We truly appreciate the thoughtful feedback and helpful suggestions, which have allowed us to further improve the clarity and scientific communication of the paper. We have addressed all comments point by point and incorporated the suggested revisions into the manuscript. We hope that the revised version meets the reviewer's expectations and further strengthens the contribution of our work.

L212: To evaluate the role of ... This sentence is grammatically incorrect.

We thank the reviewer for pointing this out. The sentence has been revised for grammatical correctness and clarity.

Fig. 1: I would suggest overlaying the observed track of Lidia in Fig. 1a. For Fig. 1b, why not overlay the observed MSLP directly over the simulated time series, instead of showing them in a subplot at the bottom right? Overlaying it directly would help readers see that the P80 member is clearly capturing a realistic RI event.

In the revised version, we now overlay the observed track in Fig. 1a and include the observed MSLP directly over the simulated time series in Fig. 1b. These changes improve readability and facilitate a clearer comparison between observations and model forecasts.

L297-298: What about the other members shown in gray-dashed lines in Fig. 1a? Many of them are even further north, which should have an even closer proximity to the trough. Why are these members not intensifying as quickly as the P80 group? I am not asking the authors to do more analysis here, but it is quite puzzling to me, and I think some clarification here would be helpful.

We appreciate the reviewer's observation. While several members track farther north, their interaction with the trough is less favorable in terms of timing, depth, and alignment of the upper-level forcing. As clarified in the revised manuscript, the  $P_{80}$  members exhibit a more optimal phasing between the trough approach and the vortex structure, which enhances synoptic-scale ascent, upper-level divergence, and midlevel moistening.

In addition, upon examining the members that recurve farther north and lie in closer proximity to the trough, we find that they experience substantially higher VWS, reduced mid- and low-level humidity, and lower SST. These environmental limitations help explain why these members do not undergo RI despite their geographic proximity to the trough. While these diagnostics are not shown explicitly in the main manuscript, this behavior is discussed in the description of the ensemble spread in Fig. 1.

Regarding L295-306, L493-502: I would suggest adding a figure or a panel showing the relative position of the trough and the TC (perhaps using the reanalysis) near the discussion of L295-306. This paragraph (L295-306) is the best place to show this information, as it allows the readers to have a clear sense of the potential importance of the trough to the RI event. Also, I found the placement of L493-502 a bit odd. I think the discussion of L493-502 is more related to the discussion in L295-306. So, overall, I would suggest merging the discussion of L493-502 with L295-306 and adding a panel (or figure) to show the TC-trough relative position clearly early on.

We thank the reviewer for this insight. We have added a panel using ERA5 reanalysis (now Fig. A2) that illustrates the relative position of the trough and the TC. Following the suggestion, we reorganized the discussion so that the interpretation formerly at L493–502 is now merged with the earlier trough–TC interaction section. This improves coherence and helps the reader connect the synoptic setup with the subsequent intensification.

The discussion about PI in L316-330: I am totally convinced that PI is not the limiting factor of the RI *of this event*, especially given that this case is clearly externally forced by the approaching trough. However, for TCs without interaction with the external environment or features, the TC would inevitably undergo RI when the environmental condition is favorable (e.g., with high PI, for example, all the idealized TC simulations in previous studies did not need external forcing and features for it to undergo RI). In those cases, having sufficiently high PI is enough to guarantee the occurrence of RI, even though the details of RI, such as the onset timing and triggering, are sensitive to the detailed vortex structure, humidity distribution at the TC inner core and boundary layer, etc. I would suggest the authors limit their discussion to *this specific case*, such as in L317-318, 322.

We agree with the reviewer and have modified the text accordingly. The revised discussion now focuses explicitly on this particular RI event, clarifying that although PI is generally sufficient for RI in isolated environments, this case was clearly dominated by external dynamical forcing from the approaching trough.

L358: Maybe also say broader and deeper? 200-hpa shows a very small difference, while 300 and 500 hpa are much clearer, so maybe mention which level you are referring to.

We have clarified the level dependence and now explicitly refer to the trough as broader and deeper at 300–500 hPa, where the differences between ensembles are most pronounced.

L362: What is a ventilation layer? Please define it clearly. I also found that the authors have a misinterpretation of what ventilation means. It would seem to me that the authors are referring to divergence. See my later comments for more details.

We have removed the term ventilation layer entirely to avoid confusion with the thermodynamic ventilation mechanism of Tang & Emanuel (2010, 2012). The revised text now refers explicitly to upper-level divergence and trough-induced flow anomalies, which correctly reflect the dynamical processes being described.

L516: Need to define V irr? Does it mean an irrotational wind vector?

We thank the reviewer for noting this omission. V\_irr has now been defined as the *irrotational wind vector* derived from the Helmholtz decomposition.

**Ventilation**: In TC dynamics, ventilation processes specifically refer to the injection of low-theta-e air from the TC environment to the TC inner core and eyewall convection, which would *suppress* the eyewall convection and intensification of TC (Tang and Emanuel 2010, 2012). There are major pathways of ventilation, including radial ventilation and downdraft ventilation (Alland et al. 2021a,b). It would seem to me that the authors are not referring to these ventilation processes. If so, please change the ventilation terminology to something else, since the current meaning and context of ventilation used here are quite contradictory to the conventional definition of ventilation (I believe that the authors think ventilation is beneficial to TC intensification).

We fully agree with the reviewer's concern. The manuscript has been revised to replace all uses of ventilation with clearer, dynamically consistent terms such as upper-level divergence, mass evacuation aloft, or trough-induced flow. Citations to Tang & Emanuel (2010, 2012) and Alland et al. (2021a,b) have been added to clarify the distinction.

The references suggested by the reviewer have also been incorporated into the revised text.

**Reference:**

Tang, B., and K. Emanuel, 2010: Midlevel Ventilation's Constraint on Tropical Cyclone Intensity. J.

Atmos. Sci., 67, 1817–1830, https://doi.org/10.1175/2010JAS3318.1.

Tang, B., and K. Emanuel, 2012: A Ventilation Index for Tropical Cyclones. Bull. Amer. Meteor. Soc., 93, 1901–1912, https://doi.org/10.1175/BAMS-D-11-00165.1.

Alland, J. J., B. H. Tang, K. L. Corbosiero, and G. H. Bryan, 2021a: Combined Effects of Midlevel Dry Air and Vertical Wind Shear on Tropical Cyclone Development. Part I: Downdraft Ventilation. J. Atmos. Sci., 78, 763–782, https://doi.org/10.1175/JAS-D-20-0054.1.

Alland, J. J., B. H. Tang, K. L. Corbosiero, and G. H. Bryan, 2021b: Combined Effects of Midlevel Dry Air and Vertical Wind Shear on Tropical Cyclone Development. Part II: Radial Ventilation. J. Atmos. Sci., 78, 783–796, https://doi.org/10.1175/JAS-D-20-0055.1.

Citation: https://doi.org/10.5194/egusphere-2025-3109-RC2